# Anti-*Candida* Activity of Extracts Containing Ellagitannins, Triterpenes and Flavonoids of *Terminalia brownii*, a Medicinal Plant Growing in Semi-Arid and Savannah Woodland in Sudan

**DOI:** 10.3390/pharmaceutics14112469

**Published:** 2022-11-15

**Authors:** Enass Y. A. Salih, Riitta Julkunen-Tiitto, Olavi Luukkanen, Pia Fyhrqvist

**Affiliations:** 1Faculty of Pharmacy, Division of Pharmaceutical Biosciences, Viikki Biocenter, University of Helsinki, P.O. Box 56, 00014 Helsinki, Finland; 2Department of Forest Products and Industries, Shambat Campus, University of Khartoum, Khartoum 13314, Sudan; 3Faculty of Science and Forestry, Department of Environmental and Biological Sciences, University of Eastern Finland, 80101 Joensuu, Finland; 4Viikki Tropical Resources Institute (VITRI), Viikki Campus, University of Helsinki, 00014 Helsinki, Finland

**Keywords:** Africa, *Terminalia brownii*, *Candida* spp., infectious diseases, HPLC-DAD and UHPLC/QTOF-MS

## Abstract

Various parts of *Terminalia brownii* (Fresen) are used in Sudanese traditional medicine against fungal infections. The present study aimed to verify these uses by investigating the anti-*Candida* activity and phytochemistry of *T. brownii* extracts. Established agar diffusion and microplate dilution methods were used for the antifungal screenings. HPLC-DAD and UHPLC/QTOF-MS were used for the chemical fingerprinting of extracts and for determination of molecular masses. Large inhibition zones and MIC values of 312 µg/mL were obtained with acetone, ethyl acetate and methanol extracts of the leaves and acetone and methanol extracts of the roots. In addition, decoctions and macerations of the leaves and stem bark showed good activity. Sixty compounds were identified from a leaf ethyl acetate extract, showing good antifungal activity. Di-, tri- and tetra-gallotannins, chebulinic acid (eutannin) and ellagitannins, including an isomer of methyl-(*S*)-flavogallonate, terflavin B and corilagin, were detected in *T. brownii* leaves for the first time. In addition, genipin, luteolin-7-*O*-glucoside, apigenin, kaempferol-4’-sulfate, myricetin-3-rhamnoside and sericic acid were also characterized. Amongst the pure compounds present in *T. brownii* leaves, apigenin and β-sitosterol gave the strongest growth inhibitory effects. From this study, it was evident that the leaf extracts of *T. brownii* have considerable anti-*Candida* activity with MIC values ranging from 312 to 2500 µg/mL.

## 1. Introduction

*Candida* spp. are eukaryotic single-celled fungi that belong to the yeast family of Saccharomycetaceae. *C. albicans* and some non-albicans *Candida* spp. cause opportunistic infections in immunocompromised individuals [1,2]. Significantly, *Candida* spp. cause systemic candidiasis (Candidemia) infection among people with immunosuppression and especially among hospitalized patients [3]. Systemic endogenous candidiasis results from the presence of *Candida* spp. in the blood circulation system or the ocular tissue (Ocular Candidiasis) [4,5]. However, *Candida albicans* is also a ubiquitous part of the human gut microbiota, and an imbalance in the microbial flora of the gut might lead to the overgrowth of yeast on the tongue, in the throat and in the colon. Moreover, in women, *Candida albicans* overgrowth can lead to vaginal yeast infection (vaginal candidiasis) [6,7,8]. Some non-*Candida albicans* species, such as *C. glabrata, C. parapsilosis, C. tropicalis* and *C. krusei*, are increasingly found externally in the oral mucosa in patients with oral thrush and especially in patients 80 years or above of age. In addition, *Candida* spp. are also a part of the normal flora on the skin and might cause (superficial) cutaneous candidiasis, especially among the elderly and people suffering from acquired immunodeficiency syndrome or skin malignancies [7,9,10,11].

So far, at least 150 *Candida* species have been recognized, of which seventeen species can induce fungal infection in humans [12,13]. Among these species, *Candida albicans* causes 80–90% of human candidiasis, followed by *C. glabrata*, *C. parapsilosis*, *C. tropicalis* and *C. krusei* [12,14], and the incidence of serious *Candida* infections is increasing [15].

Currently, there are a limited number of antifungal drugs, such as polyenes (amphotericin B, candicidin, pimaricin, trichomycin, methyl-partricin and nystatin), echinocandins (caspofungins, anidulafungin and micafungin), azoles (fluconazole, itraconazole, voriconazole, ketoconazole, miconazole, clotrimazole, ravuconazole, posaconazole), flucytosine and pyrimidine (azoxystrobin, cyprodinil, pyrimethanil and diflumetorim) [16,17]. However, the continuous emergence of resistant *Candida* spp. against the currently available antifungal drugs challenges researchers and pharmacists to urgently develop new antifungal agents from plant-derived active antifungal substances [18].

*Terminalia brownii* (*Combretaceae*) is a popular medicinal deciduous tree (growing up to 25 m) among the traditional and modern populace in wooded grassland and semi-arid woodland regions in East Africa. In African traditional medicine, *T. brownii* is used in various ways for curing and preventing several ailments and infectious diseases and among them fungal infections. Oral applications include decoctions and maceration syrups, and external applications comprise ointments, mouth washes, pastes and smoke fumigations [19,20,21,22,23,24,25]. In Sudanese vernacular Arabic, *T. brownii* is commonly known as Alshaf (الشاف), and the leaves, bark and root are traditionally used against eye infection, acne, vaginal infection, thrush causing mouth foul, skin infection and its related symptoms, such as intense itching [17,20,24,26,27,28,29]. However, still, the effective principles of antifungal *Terminalia brownii* extracts have not been studied enough.

This study aimed to verify the anti-*Candida* activity of extracts of various polarities and plant parts of *Terminalia brownii* based on its ethnopharmacological uses in Sudan. In addition, we aimed to define the chemical profiles of the active extracts and to elucidate the molecular structures of polar and non-polar agents in the extracts using high-performance liquid chromatography–diode-array detection (HPLC-DAD) and ultra-high performance liquid chromatography–quadrupole time-of-flight mass spectrometry (UHPLC/QTOF-MS).

## 2. Materials and Methods

### 2.1. Plant Identification and Authentication

Various parts of *T. brownii* were collected and dried during the period of February–March 2012 and July–August 2014 from the An-Nil Al-Azraq (Blue Nile) natural reserve forest in southeastern Sudan. Voucher specimens were authenticated by taxonomy experts El-Sheikh Abd Alla El-Sheikh (Ph.D.) and Haytham Hashim (Ph.D.) at the Faculty of Forestry, University of Khartoum, and by the article’s first author. The dried plant samples were milled in a grinder, and then the powders were checked and investigated in Plant Quarantine at Khartoum airport, Sudan (Phytosanitary certificate NO. 0005784), before being packaged in sealed paper bags and transported to the University of Helsinki, Finland, for antifungal and chemical analysis.

### 2.2. Plant Extract Preparations

#### 2.2.1. Decoctions and Macerations (Extracts Prepared from Milli-Q Water)

Cold and hot water extracts (macerations and decoctions) were prepared according to recipes customarily used in Sudanese traditional medicine. Dried, powdered samples from the various parts of *T. brownii* (20 g) were added to Milli-Q water (500 mL) that was brought to a boil, and the decoction was boiled for 5 min in a conical glass flask (500 mL *v*/*v*, Darmstadt, Germany). After boiling, the extraction was continued for 24 h by using a magnetic stirrer. For the macerations, the same procedure was made but without the boiling. In order to filter the plant material particles from the extracts, both the decoctions and macerations were centrifuged at 3000 rpm for 20 min in Eppendorf AG centrifuge tubes 5810 R (volume 50 mL, Hamburg, Germany). The supernatants were collected and filtered by percolation using a vacuum filtration technique and filter papers with a diameter of 150 mm (Schleicher & Schuell filter papers, ∅ 150 mm, Dassel, Germany). The resultant water extracts were stored at −20 °C until lyophilized in a freeze dryer for three days (HETO LyoPro 3000 Freeze Dryer, Hillerød, Denmark). The freeze-dried extracts were dissolved in methanol (50 mg/mL) for the antifungal tests.

#### 2.2.2. Hot and Cold Methanol Extracts

Hot methanolic Soxhlet extracts were obtained from 50 g of ground plant materials that were placed in a Soxhlet thimble. In a conical flask connected with the Soxhlet apparatus (Sigma-Aldrich, Schnelldorf, Germany), 500 mL of methanol was used as extraction solvent, and boiling stones were added to avoid methanol bubbling during the extraction process which was continued for 6 h. The resultant extracts were dried using, firstly, a rotary evaporator (Heidolph VV2000) for 5 h and, secondly, a freeze drier for 2–3 days (lyophilizer, HETO 154 LyoPro 3000, Hillerød, Denmark).

Cold methanol extracts were obtained using 500 mL methanol and 20 g of plant material. Overnight extractions (16–24 h) were performed at room temperature and using a magnetic stirrer (RCT, 160 digital). After extraction, samples were filtered using a Whatman filter paper (∅ 150 mm, Dassel, Germany), then dried using a rotary evaporator (Heidolph VV2000) and a lyophilizer (HETO LyoPro 3000, Hillerød, Denmark). From cold and hot methanol extracts, 50 mg/mL stock solutions were prepared for antifungal testing.

#### 2.2.3. Size Exclusion Chromatography (SEC) Using Sephadex LH-20

Size exclusion separation was performed according to [30] using the dextran-based polymer Sephadex LH-20 (Pharmacia, Uppsala, Sweden) as the stationary phase to separate high-molecular-weight tannin compounds (acetone fraction) from lower-molecular-weight compounds (ethanol fraction) in a *T. brownii* root methanolic Soxhlet extract. The plant sample was prepared for the Sephadex fractionation, so that 200 mg of the root methanolic Soxhlet extract was dissolved in 10 mL ethanol (80%, *v*/*v*) in an Eppendorf centrifuge tube (50 mL *v*/*v*) and centrifuged at 2000 RPM for 5 min (SIGMA 3-18K). Then, the upper phase was transferred to a 50 mL centrifuge tube that contained 2.5 g of Sephadex LH-20. The tube was shaken vigorously to mix the Sephadex LH-20 with the plant extract, whereafter it was centrifuged at 3000 rpm for 3 min. Then, the supernatant (the ethanol wash) was transferred to an Erlenmeyer flask. The washing of the Sephadex LH-20 with 80% EtOH was repeated until the color of the washings changed from yellow to transparent, and the resulting fractions were collected in the Erlen Meyer flask containing the ethanol fraction. Following the ethanol wash, 2 × 15 mL of acetone (70% *v*/*v*) was used as a mobile phase to release the high-molecular-weight compounds (especially tannin compounds) that were retained by the Sephadex LH-20. Rotary evaporator and liquid nitrogen were used to dry the ethanol and acetone fractions. Stock solutions of 3 mg/mL (3000 µg/mL) were prepared in methanol for antifungal testing.

#### 2.2.4. Sequential Extraction and Liquid–Liquid Fractionation

According to a method described in Salih et al. [31,32], 100 g dry powder of the leaves, stem bark, stem wood and root of *T. brownii* was extracted with 1000 mL of n-hexane or petroleum ether for 24 h. Subsequently, the marc was extracted with 1000 mL of dichloromethane or chloroform for another 24 h and finally with acetone for 24 h (for the root and root bark parts). Finally, the marc obtained from the sequential extraction process was subjected to liquid–liquid partition using equal volumes of 80% methanol and ethyl acetate which resulted in ethyl acetate and aqueous extracts. A magnetic stirrer was used overnight at room temperature to facilitate the extraction during all stages of the sequential extraction. All extracts were dried in a freeze drier (lyophilizer, HETO 175 LyoPro 3000, Denmark) for 2–3 days. The resultant extracts were dissolved in methanol at 50 mg/mL for anti-*Candida* testing.

### 2.3. Candida Strains and Assays Used for Screening Antifungal Activity

#### 2.3.1. Candida Strains

*Candida albicans* ATCC 10231, *C. glabrata* ATCC 2001 (HAMBI 2250), *C. tropicalis* ATCC 750 (HAMBI 2253) and *C. parapsilosis* ATCC 7330 (HAMBI 487) were used as model strains.

#### 2.3.2. Agar Diffusion Assays

*Terminalia brownii* leaf, stem wood, stem bark, root and root bark extracts were screened for their antifungal effects against yeasts (*Candida* spp.) using an agar diffusion method for the primary screening [33]. Before the test, Sabouraud glucose agar slants were inoculated with the yeast and incubated overnight at +35 °C. For the inoculum, a few colonies of the yeast were transferred to 2 mL of Sabouraud dextrose broth or isotonic sodium chloride (0.9% *w*/*v*). The turbidity of this suspension was measured at 625 nm using a UV-visible spectrophotometer (Pharmacia LKB-Biochrom 4060). According to the result of the turbidimetric measurement, the suspensions were diluted using Sabouraud dextrose broth or isotonic sodium chloride to reach an absorbance of 0.1 at 625 nm (≈1 × 10^8^ CFU/mL). Two hundred microliters of the fungal suspension was applied evenly on the Petri dishes (∅ = 14 cm, VWR, Finland) containing 25–30 mL of Base agar and the same volume of Sabouraud agar as a top layer. Two hundred microliters of the crude extracts (50 mg/mL), Sephadex LH-20 fractions (3 mg/mL), pure compounds (1 mg/mL) and amphotericin B (1 mg/mL, Sigma-Aldrich, USA) were applied on sterile filter paper disks (∅ 12.7 mm, Schleicher and Schuell 2668) and left to dry before applying on the agar Petri dishes. The dry disks were placed equidistantly on the Petri dishes containing fungal suspension. The Petri dishes were kept at +4 °C for one hour before incubation, whereafter the dishes were incubated at +35 °C for 48 h. The antifungal effects were measured as the diameters of the inhibition zones (IZDs). Four to six replicates of the samples were used for each experiment, and the growth inhibitory effects were reported as the mean of the replicate diameters ± standard error of means (SEM) within and between three separate experiments. Activity indexes of the extracts/fractions in relation to amphotericin B were measured as indicated in the formula below:AI (Activity index) = Inhibition zone of the plant extract/Inhibition zone of amphotericin B

In addition, the agar diffusion method was also used to estimate the approximate minimum inhibitory concentration (MIC) value for the extracts and fractions that created excess precipitation in the Sabouraud broth, which made it difficult to use the turbidimetric microplate method. For MIC assays using the agar diffusion method, 200 µL of two-fold serially diluted extracts (39–5000 µg/mL), Sephadex fractions (3000–11.72 µg/mL) and amphotericin B (0.030–1000 µg/mL) was pipetted onto sterile Whatman filter papers (∅ 12.7 mm). The lowest concentration that gave a visible growth inhibitory zone diameter (IZD) was considered the approximate MIC. The MIC values were reported as the mean of triplicates ± SEM (n = 3).

#### 2.3.3. Minimum Inhibitory Concentration (MIC) Assay Using a Turbidimetric Microplate Method

Based on the method described by the Clinical and Laboratory Standards Institute [34,35,36], a turbidimetric microdilution broth method was used to determine the MIC value for the extracts, pure compounds and amphotericin B. Extracts that gave large diameters of inhibition zones (IZDs) in the primary screenings were chosen for the microplate test. Sabouraud broth for the test was prepared in-house by adding 10 g peptone (Difco) and 20 g dextrose (Difco) to 1000 mL distilled water. Extracts were first 10-fold serially diluted from 50 mg/mL to 5 mg/mL using sterile Saboraud broth in sterile Eppendorf tubes (2 mL volume). After this dilution, two-fold dilutions were made from 5000–19.53 µg/mL. Amphotericin B and some pure compounds that were identified in *T. brownii* extracts, such as ellagic acid (E-2250, Sigma-Aldrich, Gillingham, UK), gallic acid (G-7384, Sigma-Aldrich, Shanghai, China), apigenin (Extrasynthese 286, Genay, France), corilagin (Sigma-Aldrich, Darmstadt, Germany), luteolin (Sigma-Aldrich, Darmstadt, Germany) and quercetin (Merk Art. 7546285, Darmstadt, Germany), were two-fold diluted starting from 1000–0.030 µg/mL. The Sephadex LH-20 ethanol fraction and acetone fractions of the *T. brownii* roots were two-fold serially diluted from 3000–11.72 μg/mL. A five percent volume of methanol in the wells did not affect the growth of the yeasts (*Candida* spp.). Before the test, all *Candida* spp. were cultured overnight in Sabouraud broth and incubated in an orbital shaking incubator (Stuart^®^ SI500289, London, UK) at 200 rpm, 37 °C. Using a UV-Visible Spectrophotometer (Pharmacia LKB-Biochrom 4060), the inoculum was prepared so that the final working suspension contained a cell number of 1 × 10^6^ CFU/mL. Growth control (GC) wells consisted of 100 µL suspension at 1 × 10^6^ CFU/mL and 100 µL Sabouraud broth giving a final CFU/mL in the microplate wells of 5 × 10^5^. Moreover, the test wells (GT) contained 100 μL of the two-fold diluted extracts, fractions, compounds or antibiotics and 100 μL of the diluted fungal suspension, thus also containing 5 × 10^5^ CFU/mL in the microplate wells. Since the plant extracts formed precipitation, and many of the extracts were brightly colored, controls were prepared for each plant sample, containing 100 µL of two-fold dilutions of the extract and 100 µL of broth. In addition, controls were also made for the pure compounds and antibiotics. These controls were called sample controls (SC), and their resulting absorbance at 620 nm was subtracted from the corresponding test wells containing the same plant extract, compound or antibiotic in combination with the yeast. The 96-well microplates were incubated in an orbital shaker (Stuart^®^ SI500, 289 UK) for 48 h, 200 RPM at 37 °C. After completed incubation, the optical density at 620 nm was measured using a plate reader (Victor 1420, Wallac, Finland) programmed to shake the plate immediately before the reading. The percentages of inhibition growth were calculated as the mean percentage of triplicates ± standard error of mean (SEM). The MICs were taken as the lowest concentration inhibiting more than 90% of the fungal growth. The growth of the growth control was defined as 100 percent, and the percentage growth or growth inhibition was calculated in relation to the growth control (GC) according to the below equations:
(1)$$\mathrm{Percentage fungal growth}=[(\bar{x}\;{\mathrm{GT}}_{A620}-\bar{x}{\mathrm{SC}}_{A620}/\bar{x}\;{\mathrm{GC}}_{A620})\times 100$$
Percentage inhibition of growth = 100 (% growth of the growth control, GC) − [(GT_A620_ − SC_A620_)/GC_A620_) × 100](2)
where GT_A620_ is the turbidity at 620 nm of the test well containing the plant extract, antibiotic or plant-derived compound together with the yeast; SC_A620_ is the turbidity of the sample control (the negative control for each plant extract, compound or antibiotic, containing only the extracts, compounds or antibiotic in broth); and GC_A620_ is the turbidity of the growth control at 620 nm.

Moreover, the dose dependency (linearity) of the growth inhibition percentage was plotted as the concentration of the extracts, antibiotics and pure compounds (in µg/mL) versus the resulting minimum inhibition concentration (MIC) to explain the dose–response relation (Figure 1).

### 2.4. Thin-Layer Chromatography (TLC) and DPPH Assay

Reversed-phase thin-layer chromatography aluminum-backed silica plates (RP-18F 254, 20 × 20 cm, Merck, Darmstadt, Germany) were used to fractionate the ethyl acetate extract of *T. brownii* leaves. Ellagic acid (E-2250, Sigma-Aldrich, Gillingham, UK), gallic acid (G-7384, Sigma-Aldrich, China), apigenin (Extrasynthese 286 Genay, France), corilagin (Sigma-Aldrich, Darmstadt, Germany), luteolin (Sigma-Aldrich, Darmstadt, Germany) and quercetin (Merck Art. 7546, 285 Darmstadt, Germany) were used as reference pure compounds, since many of them were previously known from *Terminalia*. A total of 20 µL of extract and compounds at 50 mg/mL and 5 mg/mL, respectively, was applied equidistantly 1.5 cm from the bottom of the TLC plate. The plate was developed in a mobile phase of methanol: water: orthophosphoric acid (50:50:1, *v*:*v*:*v*). The spots were visualized using a Camag Video documentation System (Camag Reprostar 3 TLC Visualizer) and two UV wavelengths at 366 and 254 nm.

The developed TLC plates (RP-18 F254_s_ Merck, Darmstadt, Germany) containing the ethyl acetate extract of the *T. brownii* leaves were in situ subjected to qualitative antioxidant analysis using 0.2% *w*/*v* of the 2,2-Diphenyl-1-picrylhydrazyl reagent (DPPH, Sigma-Aldrich D9132-1G, Schnelldorf, Germany). All compounds with antioxidant properties were revealed with yellow color in visible light.

### 2.5. High-Performance Liquid Chromatography (HPLC-DAD)

HPLC-DAD was used for the initial fingerprinting of a *Terminalia brownii* leaf ethyl acetate extract. The HPLC-DAD apparatus used in this work comprised Agilent Chemstation software (Water Corp., Milford, CT, USA), a reversed-phase column (Hypersil Rp C18 column, length: 60 mm; ID: 2 mm), a detector (991 PDA detector), auto-sampler and Waters 600 E pump. In 50% methanol, stock solutions of 2 mg/mL from *T. brownii* extracts were prepared, and 10 μL was injected into the HPLC device. Gradient elution was used under the pump rate of 2 mL/min, solvent (A) was 1.5% tetrahydrofuran and 0.25% orthophosphoric acid, and solvent (B) was 100 % MeOH. Different wavelengths were used at 220, 270, 280, 320 and 360 nm to identify the various peaks. Using Agilent Chemstation software, UV-visible-absorption maxima spectra were recorded between 200 and 400 nm. Unknown and known peaks were studied according to the database of natural compounds available in the computer library and the literature.

### 2.6. Mass Spectrometry (UHPLC/QTOF-MS)

A method described by Taulavuori et al. [37] was employed to profile the polyphenolic and phenolic compounds and other chemical constituents in a *T. brownii* leaf ethyl acetate extract. For measuring the molecular weight of the detected compounds, an ultra-high-performance liquid chromatograph (UHPLC-DAD, Model 1200 Agilent Technologies)-JETSTREAM consisting of a reversed-phase column C_18_ (2.1 × 60 mm, 1.7 μm, Agilent technologies) was connected with QTOF-MS Agilent Technologies (Model 6340). Negative-ion mode [M-H]^−^ and mass-to-charge ratio ranging between *m*/*z* 100 and 2000 were employed. The mobile phase comprised solvent A: 1.5% tetrahydrofuran and 0.25% acetic acid in ionized water and solvent B: 100% methanol. Gradient-elution chromatography was set as follows: from 0 to 1.5 min, B 0%, from 1.5 to 3 min, 0 to 15% B, from 3 to 6 min, 10 to 30% B, from 6 to 12 min, 30 to 50% B, from 12 to 20 min, 50 to 100% B, and from 20 to 22 min, 100 to 0% B. The accuracy of the measured molecular weights (M_measured_) compared to the compound calculated molecular weight (M_calculated_) of the identified molecules were set as parts per million (ppm) according to the below formula:PPM (ppm, parts per million mass error) mass accuracy = (M_measured_ − M_calculated_) × 10^6^/M_calculate_

M_measured_ = measured mass in QTOF-MS; M_calculated_ = exact calculated mass according to the CAS molecular formula of the identified compound. The molecular weight of the hydrogen atom (1.0078) was subtracted from all the calculated masses to achieve the negative ions [M-H]^−^. Exact molecular weights were calculated according to monoisotopic masses for the carbon atom of 12.000, the oxygen atom of 15.9949 and the hydrogen atom of 1.0078.

## 3. Results

### 3.1. Effects of the Extracts against Yeasts

Our primary screening results using the agar diffusion method show that almost all forty-five extracts of the various parts of *T. brownii* inhibit the growth of the tested *Candida* spp. (Table 1). The most effective extracts gave an MIC of 312 µg/mL, compared to 3.9–62.5 µg/mL for amphotericin B (Table 2), and the inhibitory effects of the extracts were often concentration dependent (Figure 1). In particular, many of the leaf extracts gave good activities, with sixteen extracts showing MIC values under 1000 µg/mL and four extracts showing an MIC value of 312 µg/mL. These extracts were the acetone, ethyl acetate, aqueous and a hot methanol extract of the leaves (Table 2). Moreover, of these extracts, the leaf acetone extract was the most effective and gave an MIC of 312 µg/mL against *C. parapsilosis*, *C. albicans* and *C. glabrata*. The acetone extract of the leaves also gave large inhibition zone diameters (IZDs) of 28–38.6 mm against the mentioned yeast species, and this result correlates well with the MIC results (Table 1). Likewise, acetone and hot methanol Soxhlet extracts of the leaves presented MIC values of 625 µg/mL and large IZDs of 29.83 and 30.90 mm, respectively, against the growth of *C. tropicalis* (Table 1 and Table 2). Notably, hot and cold water extracts of the leaves prepared in the traditional way inhibited the growth of *C. albicans* and *C. parapsilosis* at an MIC value of 625 µg/mL and with inhibition zone diameters ranging from 26 to 32 mm (Table 1 and Table 2).

Extracts of the whole root resulted in the second best growth inhibitory effects after the leaf extracts, and altogether fourteen extracts had an MIC lower than 1000 µg/mL. Polar extracts of the root, such as the acetone, cold methanol and methanol Soxhlet extract, showed an MIC value of 312 µg/mL against *C. glabrata* and large inhibition zone diameters (IZD 40–40.6 mm). Good antifungal activity was also shown by the root ethyl acetate and aqueous fractions that gave an MIC value of 625 µg/mL against *C. glabrata* and *C. albicans* (Table 2). The root decoction was active against *C. albicans*, *C. parapsilosis* and *C. glabrata* (MIC 1250 µg/mL) and was more active than the maceration.

Sephadex LH-20 fractionation of the hot methanol extract of the roots resulted in increased antifungal activity of the obtained fractions against *C. parapsilosis*. The ethanol and acetone fractions (acetone wash) showed MIC values of 93.75 and 375 µg/mL, respectively, compared to 625 µg/mL for the hot methanol extract. Moreover, the ethanol fraction (ethanol wash) also displayed good activity against *C. albicans* and *C. tropicalis* with an MIC value of 187.50 µg/mL compared to MIC values of 312 and 1250 µg/mL for the hot methanol extract (Table 2).

When compared to the roots and the leaf extracts, the stem bark and stem wood had a smaller number of extracts showing an MIC value of 312 µg/mL, namely the stem wood methanol Soxhlet extract against *C. albicans* and the stem bark hot and cold methanol extracts against *C. glabrata*. Notably, decoctions and macerations of the stem wood and stem bark were active against *C. albicans*, *C. parapsilosis* and *C. glabrata* with MIC values of 625–1250 µg/mL, the stem wood being more active than the stem bark.

In general, the non-polar extracts of hexane and dichloromethane showed weak to moderate activities against all the tested yeast strains. However, hexane extracts of the stem wood and roots showed the best results among the non-polar extracts (Table 1 and Table 2). As shown in Table 1, the hexane extracts of the leaves, stem wood, root and root bark gave large inhibition zone diameters against *C. tropicalis,* ranging between 26 and 29 mm. Dichloromethane extracts of the root, stem wood and stem bark displayed larger inhibition zone diameters (IZDs) compared to the root bark and leaf dichloromethane extracts. Moreover, no growth inhibition was shown by hexane extracts of the leaves, root and root bark against *C. albicans* (Table 1).

### 3.2. Effects of Pure Compounds Present in the T. brownii Extracts against Candida *spp*. Growth

The results of some commercially available pure compounds that were either present in the antifungal extracts of *T. brownii* or are known to occur as aglycones in *Terminalia* spp. (such as quercetin) are presented in Table 2. Apigenin, which was found to be present in the ethyl acetate extract of *T. brownii* leaves in this investigation (Table 3), and the phytosterol β-sitosterol, which we have found earlier in n-hexane extracts of the stem bark and wood [25], were the most active compounds against *C. parapsilosis* showing MIC values of 125 µg/mL. In addition, apigenin inhibited the growth of *C. albicans* and *C. glabrata* at an MIC of 500 µg/mL, and β-sitosterol inhibited *C. glabrata* with an MIC value of 250 µg/mL. Quercetin, of which we have found the di-glycoside and galloyl-glycoside derivatives in the stem bark and wood of *T. brownii* [38], gave the best antifungal effects against *C. albicans* and *C. glabrata* (MIC 250 µg/mL) (Table 2 and Figure 1). Luteolin was mildly active against *C. tropicalis* and *C. parapsilosis* (MIC 500 µg/mL) and *C. albicans* (MIC 1000 µg/mL) (Table 2). Moreover, β-sitosterol and stigmasterol showed mild growth inhibitory effects against *C. albicans* and *C. tropicalis* (Table 2 and Figure 1).

Friedelin, a pentacyclic triterpene that we have found previously in a hexane extract of the stem bark of *T. brownii* [25] showed an MIC value of 500 µg/mL against *C. parapsilosis* and *C. tropicalis* (Table 2).

Ellagic acid, often present as a result of the degradation of ellagitannins in alcohol extracts of *Terminalia* or as a sugar derivative, resulted in an MIC value of 500 µg/mL against *C. glabrata* and *C. parapsilosis*. Gallic acid showed only a slight anti-*Candida* effect with an MIC value at 1000 µg/mL against all tested yeasts (Table 2).

Corilagin, a small molecular weight ellagitannin that was found in the leaf extract of *T. brownii* in this investigation, gave MIC values of 500 and 1000 µg/mL, respectively, against *C. tropicalis* and *C. albicans* (Table 2).

### 3.3. Detection of Compounds with Antioxidant Effects

An ethyl acetate extract of *T. brownii* leaves that gave promising anti-*Candida* effects in this investigation was chosen to assess the quantity and quality of compounds with antioxidant effects. As shown in Figure 2, altogether eighteen spots were visible on the TLC plates at 254 and 366 nm. Many of these compounds showed strong antioxidative effects. Especially the standard compounds gallic acid, corilagin and ellagic acid, which are also present in the extract, showed strong antioxidant effects as they turned intensively light yellow after spraying with the DPPH reagent (Figure 2). In addition, ellagitannins and flavonoids that occur in the extract give a strong antioxidant effect (Figure 2).

### 3.4. Phytochemistry of an Ethyl Acetate Extract of the Leaves

An ethyl acetate extract of the leaves of *T. brownii* was chosen for phytochemical analysis since the extract showed a good anti-*Candida* profile in this investigation, and moreover, the leaves have not earlier been studied in depth for their chemical constituents. The results from our extract profiling and the molecular masses of the characterized compounds are shown in Figure 3 and Table 3. All molecular ions were obtained using negative mode, which is particularly useful for polyphenols, such as ellagitannins (e.g., Pfundstein et al., 2010 [39]). Our results show that the ethyl acetate extract of the leaves is rich in polyphenolic constituents, especially hydrolyzable tannins, such as ellagitannins and gallotannins and their isomers as well as flavonoids and a high number of unknown compounds, including the iridoid glucoside genipin.

In total, the phytochemical analysis resulted in the detection of sixty compounds, and among them, twenty-nine were gallotannins and fifteen ellagitannins (Table 3).

A number of ellagitannins that have not previously been characterized in *T. brownii* were found in this screening based on their UV absorbance maxima data, retention times and molecular masses (Table 3, Figure 3, Figure 4 and Figure 5). For example, an ellagitannin (**12**) at tR 13.92 min (HPLC-DAD) and tR 4.529 (UHPLC-DAD) gave a molecular ion [M-H]^−^ at *m*/*z* 617.0164 and was the major compound in the ethyl acetate fraction (peak area 15.46 %). In addition, two other unknown ellagitannins were characterized: compound (**8**) at tR 10.19 min (UHPLC-DAD) and 3.363 min (UHPLC-DAD) with a molecular ion at *m*/*z* 541.0291, and compound (**5**) was detected due to its UV absorption maxima, with three absorbance peaks, typical for ellagitannins (Table 3, Figure 4). Moreover, four previously known ellagitannins were characterized: terflavin B (**10**) at tR 11.30 min (UHPLC-DAD) and 3.546 min (UHPLC-DAD) showing a molecular ion [M-H]^−^ at *m*/*z* 783.0701, an isomer of methyl-(*S*)-flavogallonate (**6**) at tR 8.92 min (HPLC-DAD) (*m*/*z* 483.0811) and corilagin (**15**) at tR 16.03 min (HPLC-DAD) (*m*/*z* 633.0756) (Table 3). Chebulinic acid (1,3,6-tri-*O*-galloyl-2,4-*O*-chebuloyl-β-*D*-glucose) (**27**), also called eutannin, at tR 22.8 min showed a molecular ion at *m*/*z* 955.079. Chebulinic acid was characterized for the first time in the leaves of *T. brownii* (Figure 4 and Figure 5, Table 3). In addition to the ellagitannins, two ellagic acid derivatives that are thought to be derived from ellagitannins were characterized. The first one was di-methyl ellagic acid glucoside (**23**) (*m*/*z* 491.0843), and the other was identified based on the UV_λ_ maxima absorbance spectrum (**31**) (Table 3).

Twenty-nine gallotannins and their derivatives were characterized, and many of them for the first time. Among the gallotannins were 1,3-di-galloyl-β-*D*-glucose (**4**) (*m*/*z* 483.0796), 1,6-di-galloyl-β-*D*-glucose (**9**) (*m*/*z* 483.0808) and 2,4-di-*O*-galloyl-1,5-anhydro-*D*-glucitol (Maplexin D) (**19**) (*m*/*z* 467.0802). In addition, two tri-galloylated gallotannins, 1,2,3-tri-*O*-galloyol-β-*D*-glucose (**13**) (*m*/*z* 635.0889) and 3,4,6-tri-*O*-galloyl-β-*D*-glucose (**18**) (*m*/*z* 635.0904) were characterized from the leaves of *T. brownii*, and the structures were compared with previous analytical work of galloylated hydrolyzable tannins [40,41]. Moreover, two other gallotannins were eluted very close to each other and not completely separated (peak **21a** and **b)** (Figure 3). The other one (**21a**) was identified as 1,2,3,6-tetra-*O*-galloyl-β-*D*-glucose (*m*/*z* 787.0872).

Nine flavonoids were identified in the leaf extract based on their UV_λ_ absorption maxima and retention times. UV_λ_ absorption maxima are shown in Figure 4 and Table 3. Among these, three flavonoids could be identified also due to their molecular masses. These flavonoids were the flavone-based luteolin-7-*O*-glucoside (**16**) (*m*/*z* 447.0945), apigenin (**20**) (*m*/*z* 269.0455) and myricetin-3-rhamnoside (**24**) (*m*/*z* 463.3798). In addition, kaempferol-4’-sulfate (**22**) was characterized at *m*/*z* 365.2928, according to the reported NMR analysis for a *T brownii* leaf extract [29]. By comparison with literature data reported by [42,43] on the presence of genkwanin in *Combretum erythrophyllum*, a species related to *Terminalia*, we also tentatively identified genkwanin (apigenin 7-*O*-methyl ether) in the leaf ethyl acetate extract of *T. brownii* at tR 39 min (compound **33**)**.**

At the end of the HPLC chromatogram (tR 30.878 min), we detected the monoterpenoid genipin (**29**), an aglycone that is derived from the iridoid glycoside geniposide. Genipin showed an [M-H]^−^ molecular ion at *m*/*z* 225.1504. Our result is in an agreement with the previous reported data for detecting genipin at the UV_λ_ absorbance maxima of 238 nm and 240 nm in acetic acid 1% (*v*/*v*) and methanol 45% (*v*/*v*) [44,45]. In addition, the triterpenoid sericic acid (**30**) with a [M-H]^−^ molecular ion at *m*/*z* 503.3379 was identified at tR 35.44 min in HPLC-DAD. Two unknown non-polar compounds of (**32**) are also identified at retention times in HPLC-DAD ranging from 38.72 to 39.01 min, giving mass to charge 293.1759 and 283.1920 in QTOF-MS, respectively.

## 4. Discussion

### 4.1. Anti-Candida Effects of T. brownii Extracts in Relation to Other Research on Anti-Candida Effects of Terminalia Species

Our results indicate that especially polar extracts of the leaves, stem bark and stem wood of *T. brownii* are antifungal, and our results are in agreement with other authors [17,21,22,29,38,46,47,48,49,50]. In addition, we found that a number of root extracts inhibited the growth of yeasts. To the best of our knowledge, the root part of *T. brownii* has not been tested before for its anti-*Candida* effects. However, most authors have tested *T. brownii* primarily against *C. albicans*, whereas other *Candida* species, such as the increasingly significant human pathogens *C. glabrata*, *C. parapsilosis* and *C. tropicalis* [51], have not been included. Thus, we report for the first time that *T. brownii* extracts of the leaves, stem bark, stem wood, roots and root bark are active against three non-*albicans*
*Candida* spp.: *C glabrata*, *C. tropicalis* and *C. parapsilosis*. For example, we found that in addition to giving good activities against *C. albicans*, the acetone, ethyl acetate, aqueous and methanol Soxhlet extracts of the leaves showed promising growth inhibitory effects against *C. glabrata* and *C. parapsilosis*, as judged by the large number of extracts showing the lowest MIC value detected in our tests, 312 µg/mL (Table 2). Moreover, the MIC of 312 µg/mL correlated well with large inhibition zones for the acetone, methanolic Soxhlet and hot water extracts of the leaves (Table 1 and Table 2). The ethyl acetate extract of the leaves of *T. brownii* is rich in ellagitannins, ellagic acid derivatives and gallotannins, and among the ellagitannins, terflavin B, corilagin and chebulinic acid could contribute to the antifungal effects of the extract (Figure 3). 

In this study we found that root extracts of *T. brownii* give promising anti-*Candida* activity with MIC values between 312 and 625 µg/mL for the most effective extracts. Especially good growth inhibitory effects were shown against *C. glabrata* and in some cases also against *C. albicans*, and the hydrophilic extracts were in general more active than the hydrophobic ones. Our results are in agreement with [50] who found that ethyl acetate, ethanol and water extracts of the roots of *T. brownii* give good growth inhibitory effects against *C. albicans*. In addition, we found that the by far best antifungal effects were obtained after Sephadex LH-20 purification of a root methanol Soxhlet extract, and the ethanol wash gave better effects than the acetone wash (lowest MIC 93.75 versus 375 µg/mL against *C. parapsilosis*) and 187 µg/mL for the ethanol wash against *C. tropicalis* and *C. albicans* (Table 2). In our previous paper Salih et al. [25], we hypothesized that since the Sephadex LH-20 fractions of a root methanol extract differed from each other both qualitatively and quantitatively to their polyphenol composition, and since the ethanol wash contained more of certain compounds, such as an isomer of methyl-(*S*)-flavogallonate and ellagic acid, these compounds might contribute to its superior antifungal effects compared to the acetone wash. Moreover, generally speaking, especially the roots of plants are known to contain a high number and variety of antimicrobial chemical constituents compared to other plant parts as we have reported earlier [24].

Our results on the antifungal effects of extracts of *T. brownii* are in accordance with several other investigations on *Terminalia* spp. against various opportunistic yeasts such as *T. arjuna* [52], *T. catappa* [53] bark and *T. chebula* fruit [54], that revealed antifungal activity against *Candida albicans*, *C. glabrata*, *C. krusei* and *C. tropicalis* [55,56].

In Sudan, as in many other countries of Africa, *T. brownii* is used as a mouthwash for the treatment of fungal infections, including oral thrush resulting from *Candida* spp. overgrowth. The traditional preparations include hot water decoctions and macerations [26]. According to our study, decoctions and macerations of the leaves, stem wood and roots gave good growth inhibitory effects against the *Candida* spp. with the lowest MIC of 625 µg/mL and concentration-dependent effects (Table 2, Figure 1) and thus support the traditional medicinal use of this plant for the treatment of fungal infections, including skin infections, diarrhea and oral thrush.

### 4.2. Antioxidant Potential of a Terminalia brownii Ethyl Acetate Extract of Leaves

We found that the *T. brownii* ethyl acetate extract of the leaves contained various antioxidant polar compounds, among them flavonoids, tannins, gallic acid and ellagic acid (Figure 2). Earlier, we have seen that the phenolic compounds present in the methanol extracts of many species of *Terminalia* are also present in the water extracts, e.g., decoctions and macerations [35]. Therefore, we assume that the phenolic compounds present in the ethyl acetate extract of *T. brownii* would be present also in the traditional preparations, such as the polar decoctions and macerations used for the treatment of fungal infections. These polyphenols play an important role as antioxidant agents to promote the host immune system and might as well play roles as pro-oxidant agents to induce oxidative stress in *Candida* spp., due to the inhibition of the activity of the antioxidant enzyme system, decreasing the activity of all other antioxidant enzymes, except catalase, and thus leading to damaged yeast cells [57]. 

Our finding that the leaves of *T. brownii* are rich in antioxidant compounds is in agreement with a previous report that *T. brownii* leaves contain antioxidant compounds [58,59]. These results warrant further consideration to quantify the antioxidant capacity of the pure compounds in relation to a combination of compounds (fractions and extracts) per one TLC spot.

### 4.3. Identified Compounds in a Leaf Ethyl Acetate Extract of T. brownii and Their Suggested Influence on the Growth of Candida *spp.*

#### 4.3.1. Ellagitannins and Ellagic Acid Derivatives

*Terminalia brownii* is rich in ellagi– and gallotannins, both according to our previous results on hydrolyzable tannins in the root part [24,25] and according to our present research on the leaves (Figure 3 and Figure 4). In our present research, we found that the leaf part of *T. brownii* contains chebulinic acid, terflavin B and an unknown ellagitannin at tR 13.9 min and with a [M-H]^−^ ion of 617.0164 as well as a number of other ellagitannins. In addition, we found that corilagin was present in the leaves. To the best of our knowledge, chebulinic acid (eutannin) or 1,3,6-tri-*O*-galloyl-2,4-*O*-chebuloyl-β-*D*-glucose has not been reported previously in the leaves of *Terminalia* spp. and is not previously known from *T. brownii*. Chebulinic acid has been characterized earlier from the fruits of *T. chebula* [39,60,61,62]. The ellagitannin at tR 13.9 min was present in a high concentration (15% peak area), whereas terflavin B had a peak area of 5% and chebulinic acid 10%. Thus, these ellagitannins could have a major impact on the antifungal effects of the ethyl acetate extract of the leaves. Of these ellagitannins, chebulinic acid, isolated from *Terminalia chebula*, was found to be active against *C. albicans* at an MIC concentration of 0.025 µg/mL [62]. According to our knowledge, terflavin B and methyl-(*S*)-flavogallonate have not been tested for their growth inhibitory activity against yeasts. The high concentration of terflavin B and chebulinic acid that we found in the leaves of *T. brownii* warrants further testing on the anti-*Candida* effects of these ellagitannins, since chebulinic acid was found to be active against *C. albicans* [62], and moreover, they might have an important contribution to the overall antifungal effect of the ethyl acetate leaf extract. Among the other ellagitannins that we found in *T. brownii*, corilagin gave only a mild antifungal effect against *C. tropicalis* in our screenings (MIC 500 µg/mL) and was not active at 1000 µg/mL against the other *Candida* strains. This finding is in accordance with the study of [63] who also found that corilagin gave mild growth inhibitory effects against *C. tropicalis*. However, oppositely to us, Latté et al. [64] demonstrated a strong antifungal effect for corilagin with an MIC at 31 µg/mL against *Candida glabrata* (ATCC 90876 and ATCC 90030) and *Cryptococcus neoformans* (ATCC 34544 and DSM 70219). This might be due to the use of different strains compared to ours and differing screening methods. For example, a small inoculum of 1 × 10^2^ CFU/mL was used for the screenings.

We assume that the traditional medicinal preparations of *T. brownii*, such as the maceration and decoction of the leaves, contain the same ellagitannins that we found in the ethyl acetate extract. Therefore, these ellagitannins might contribute to the antifungal properties of the macerations and decoctions and especially if used topically or as mouth washes, since the ellagitannins are not needed to be metabolized to reach the skin and the mouth.

Pure ellagic acid was not present in the leaf ethyl acetate extract of *T. brownii* but was present as the methylated and glucosylated derivative, di-methyl ellagic acid glucoside. However, in our earlier research, we found ellagic acid in a root extract of *T. brownii* [24]. In our study, due to the difficulty of obtaining di-methyl ellagic acid glucoside as a standard compound, we used ellagic acid as a model compound for ellagic acid compounds. We found that ellagic acid gives a mild growth inhibitory effect against *C. glabrata* and *C. parapsilosis* with an MIC value of 500 µg/mL. This result is supported by Silva Junior et al. [65] who reported on a broad-spectrum growth inhibitory activity for ellagic acid against many *Candida* spp., including *C. albicans* and *C. parapsilosis* (MIC 500 and ˃1000 µg/mL, respectively). Moreover, Sampaio et al. [66] indicated an anti-biofilm effect for ellagic acid derivatives at a concentration of 250 µg/mL against *C. albicans*. The antifungal mechanism of action for ellagic acid and its derivatives, such as di-methyl ellagic acid glucoside (**23**) that we found in the leaves of *T. brownii*, is probably implicated via the prevention of the biosynthesis of ergosterol, an important compound in the *Candida* plasma membrane [67]. Thus, ellagic acid and its derivatives could be interesting targets for new antifungal drugs from a sustainable natural source, such as *T. brownii*, to be used instead of or as a complement to the conventional antifungal drugs, such as azoles, polyenes and echinocandins [68,69].

#### 4.3.2. Gallotannins

Similarly to Pfundstein et al. [39], who identified various gallotannins from *Terminalia chebula* and *T. horrida*, we also described the occurrence of a high number of gallotannins in *T. brownii* leaves, including di-, tri- and tetra-galloylglucose (Table 3). Gallotannins have been found to possess antifungal activity; hepta-*O*-galloylglucose isolated from mango kernels was active against *Aspergillus niger* and *Penicillium* spp. with an MIC value of 3300 mg/mL [70]. The antifungal mechanism of action of the gallotannins involves disruption of the fungal membrane integrity and respiration [70]. The large number of gallotannins found in ethyl acetate and other hydrophilic extracts of *T. brownii* leaves in this present research might contribute significantly to the growth inhibitory effects of theses extracts against the tested yeasts. Therefore, the tannin-rich water decoctions and macerations of *T. brownii* leaves (and other parts) that are used by traditional healers might also be active against *Candida* spp. due to the presence of a high number of ellagi- and gallotannins in these preparations. Moreover, as an additional health effect of tannins, it has been claimed that tannin extracts do not inhibit the growth of health-promoting probiotic microorganisms that are present in the human gut [71,72].

#### 4.3.3. Gallic Acid and Protocatechuic Acid

We found that *T. brownii* leaves contain gallic acid and protocatechuic acid. Gallic acid showed antifungal effects against *C. albicans*, *C. glabrata* and *C. parapsilosis* with an MIC value of 1000 µg/mL [73]. In addition, we found that protocatechuic acid, which is a prenylated phenolic acid, inhibited the growth of *C. tropicalis* and *C. albicans* at an MIC of 400 µg/mL and 500 µg/mL, respectively. However, recently, Fifere et al. [74] reported that protocatechuic acid at concentrations ranging from 50 to 200 µg/mL does not affect the growth of *C. albicans* ATCC 10231. Likewise, protocatechuic acid and gallic acid showed MIC values of ˃2000 µg/mL against the food-contaminating yeasts of *Saccharomyces cerevisiae*, *Debaryomyces hansenii* and *Schizosaccharomyces pombe* and the Ascomycete, *Pichia anomala* [75].

#### 4.3.4. Flavonoids

Our results reveal that *T. brownii* leaves contain a large variety of flavonoids, including luteolin-7-glucoside, apigenin, kaempferol-4′-sulfate, myricetin-3-rhamnoside and genkwanin (syn. apigenin-7-*O*-methyl-ether). Concerning the unusual sulfate-containing flavonoid, kaempferol-4′-sulfate (**22**), it has been reported previously in *T. brownii* leaves and was found to contribute to a low MIC value of a fraction isolated from *T. brownii* against bacteria and fungi (MIC 4–6 µg/mL) [29]. Kaempferol-4′-sulfate is a phytoanticipin and belongs thus to those flavonoids that are present in the plant already before fungal infection, as a possible constitutive defense [29,76]. Other kaempferol derivatives, such as kaempferol 3,7-di-sodium sulfate and kaempferol 7-*O*-methyl-3-sulfate, were found in other plant families as in *Frankenia laevis* (Frankeniaceae) and *Argyreia speciosa* (Convolvulaceae), respectively [77]. The sulfur moiety in kaempferol-4′-sulfate might interfere with *Candida* sulfur metabolism, such as the cysteine and methionine synthase enzymes, due to its hydrophilic character and inhibit the virulence factors of *Candida* spp. [78,79]. Moreover, it has been claimed that sulfur moieties (in flavonoids and other compounds) might inhibit the formation of polysaccharides, DNA, RNA and protein in the fungal cells [79,80]. Furthermore, other kaempferol derivatives, such as kaempferol 3-*O*-rutinoside showed a moderate MIC value of ˃500 µg/mL against *C. albicans* and *C. parapsilosis* according to Nascimento et al. [81].

Many authors have found that a glycosyl moiety in flavonoids, such as rhamnose and glucose in myricetin-3-rhamnoside (**24**) and luteolin-7-*O*-glucoside (**16**), of which both flavonoids were found in this study in *T. brownii* leaf extracts, could assist the entry of the flavonoid compound into the *Candida* cell via porins, where hydrolyzation of the sugar leads to the release of the flavonoid aglycone that can disrupt the *Candida* membrane [82,83,84]. Moreover, for example, in the mango fruit peel, many flavonoids are concealed as inactive glycosides but are released as aglycones, for example, after fungal attack [84].

Moreover, regarding the structural requirements for the antifungal potency of flavonoids, the substitution of a hydroxyl group in position 7 of the A-ring, such as in myricetin, might play a major role in the anti-*Candida* potential of flavonoids [85]. Previously, Martins et al. [86] reported that myricetin-3-*O*-rhamnoside had an MIC of 93.8 µg/mL against the growth of *C. albicans* and *C. tropicalis*. Therefore, the glycosidic and hydroxylated flavonoids that we found in the ethyl acetate extract of *T. brownii* leaves could play a major role in the antifungal activities of this extract.

We found that apigenin, luteolin and quercetin, which are aglycone flavonoids present in *T. brownii* extracts, were antifungal against *Candida* spp. with MIC values ranging from 125 to 500 µg/mL. Apigenin was most effective (**20**) against the growth of *C. parapsilosis* with an MIC of 125 µg/mL (Table 2). Overall, the antifungal activity of these flavonoids was found to be moderate to relatively low in this present investigation compared to the root and leaf crude extracts of *Terminalia brownii*. Previously, apigenin was found to inhibit the growth of *C. albicans* with an MIC of 100 µg/mL [87,88]. Recently, Ivanov et al. [89] and Aboody and Mickymaray [7] found that extracts containing apigenin inhibit biofilm formation in *C. albicans* as well as inhibit the efflux pump function in fungal cells and thus induce cell death in fungi.

In correlation to our study, various flavonoids were detected earlier in *Terminalia* spp.: myricetin and kaempferol derivatives from the bark, fruit and wood extracts of *T. catappa* [90]; apigenin-*O*-galloyl-β-*D*-glucopyranoside, vitexin (apigenin glucoside) and isovitexin (apigenin-6-*C*-glucoside) in the leaves of *T. catappa* [91,92]; quercetin, kaempferol and luteolin in the leaves of *T. arjuna* [93,94]. In addition, genkwanin (apigenin-7-*O*-methyl ether) was isolated from *C. erythrophyllum* [42,43] and gave an MIC value of ˃100 µg/mL against *Aspergillus niger* [42].

#### 4.3.5. Monoterpenoids

In this study, we found that the iridoid derivative of genipin (**29**) was present in the leaves of *T. brownii*. Genipin was identified previously from the stem bark of *T. brownii* [95] and possessed an ex vivo growth inhibitory effect against *C. albicans* at an MIC value of 625 µg/mL [96]. Furthermore, genipin was found also in many other medicinal plants, such as the Rubiaceae species of *Genipa americana* and *Gardenia jasminoides* [44,97]. The antifungal efficiency of genipin refers to its potential as a strong natural and efficient cross-linker (bioconjugation) targeting the chitosan and protein functional groups in fungal cells, as well as attributed to its resistance properties to enzymatic digestion [96,98,99].

#### 4.3.6. Triterpenes, Sterols and Fatty Acids

A common overall trend has been reported that non-polar plant extracts of *Terminalia* spp. are less antifungal compared with polar extracts [100]. In agreement with this trend, we found that non-polar extracts of *T. brownii* were less antifungal than polar extracts (Table 1 and Table 2). However, at higher concentrations, the non-polar extracts are active, and according to our results, hexane and dichloromethane extracts from various parts of *T. brownii* gave MIC values between 1250 and 5000 µg/mL against the *Candida* species. These activities could be related to the presence of triterpenes, sterols and fatty acid constituents, which we reported in this study (Table 3) and in our previous publication [25]. Previously, Opiyo et al. 2011 [47] and Machumi et al. 2013 [22] found sericic acid, friedelin, β-sitosterol and stigmasterol in the stem bark extracts of *T. brownii*. However, sericic acid has not been reported before in leaf extracts of *T. brownii*, but it has been characterized in the root bark of other *Terminalia* spp., such as *T. glaucescens* and *T. sericea* root bark [101,102].

In our study, it was found that β-sitosterol, stigmasterol and friedelin showed MIC values ranging from 125 to 1000 µg/mL against the growth of the tested *Candida* (Table 2), with β-sitosterol being most effective against *C. parapsilosis* (MIC 125 µg/mL). Our MIC result for β-sitosterol against *C. albicans* is higher (1000 µg/mL) than those obtained by Akbar et al. [103] who found that β-sitosterol gave an MIC of 182 µg/mL and minimum fungicide effect (MFC) at a concentration of 50 µg/mL against *Candida albicans* [104]. The difference in MIC values could be due to the use of different strains of *C. albicans*.

Friedelin and stigmasterol displayed MIC values of 500–1000 µg/mL in our study. In an earlier study, friedelin and stigmasterol gave MIC values of ˃200 µg/mL and 100 µg/mL, respectively, against *C. albicans*, and stigmasterol gave an inhibition zone of 25 mm against *C. albicans* [105,106]. β-sitosterol, stigmasterol and friedelin that were present in *Cassia fistula* oil were found to inhibit the ergosterol biosynthesis in the *Candida* cell membrane [107]. Therefore, the anti-*Candida* activity exhibited by *T. brownii* hexane and dichloromethane extracts in our study could be attributed to the presence of specific sterol, triterpene and fatty acid constituents that have been demonstrated in this study and in our previous work [25]. It is possible that the various compounds in the extracts could act synergistically with each other to enhance the effects of the individual compounds.

## 5. Conclusions

Our results together with those by other authors indicate that *T. brownii* leaves, stem bark, wood, root and root bark contain antifungal compounds with a wide range of polarities and warrant further studies in the quest to find new single antifungal compounds and antifungal compound combinations from this species.

The observed anti-*Candida* activity of many extracts obtained from *T. brownii* seems to be due to the fact that these extracts contain a large variety of phenolic compounds such as flavonoids and tannins as well as iridoids and terpenes. The detected ellagitannins, gallotannins and terpenes, and among them genipin and chebulinic acid in *T. brownii* leaf extracts, are reported for the first time. The study validates and confirms the traditional uses of *T. brownii* crude extracts to treat various fungal infections. *T. brownii* contains a considerable variety of antifungal compounds, and various extracts and compounds indicate promising growth inhibitory effects against *Candida* spp. Possible synergistic actions of various compounds in decoctions and macerations might further confirm the customary use of these preparations in Sudan against *Candida* spp. infections.

## Figures and Tables

**Figure 1 pharmaceutics-14-02469-f001:**
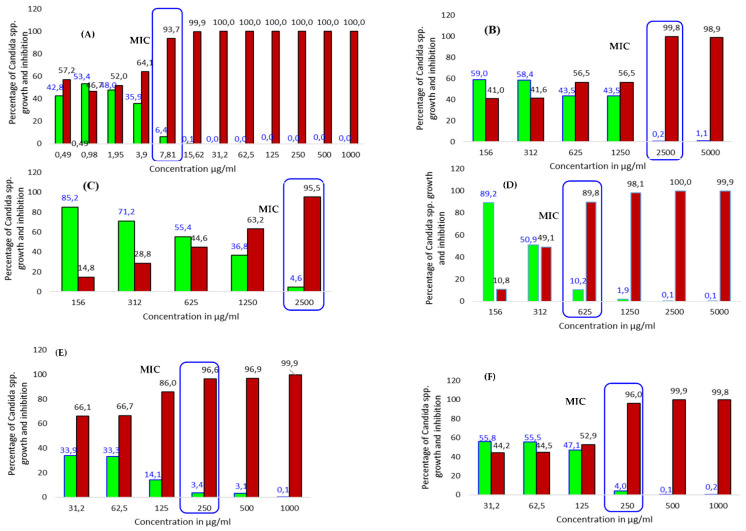
Anti-*Candida* activity of *Terminalia brownii* extracts, pure compounds and amphotericin B. (**A**) Amphotericin B against *Candida tropicalis*; (**B**) decoction of the leaves against *C. glabrata*; (**C**) maceration of the root bark against *C. tropicalis*; (**D**) decoction of the leaves against *C. albicans*; (**E**) quercetin against *C. albicans*; (**F**) sitosterol against *C. glabrata*; 
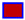
 percentage of growth inhibition; 
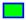
 percentage of growth.

**Figure 2 pharmaceutics-14-02469-f002:**
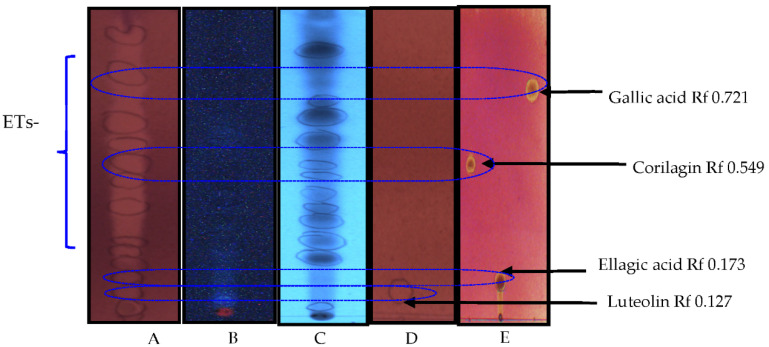
Thin-layer chromatograms (RP 18-TLC) of an ethyl acetate extract (EtOAc) of the leaves of *Terminalia brownii* (**A**–**C**) and standard compounds (**D**,**E**). The chromatograms were imaged after treatment with the DPPH reagent (**A**,**D**,**E**) at 366 nm (**B**); at 254 nm (**C**). Rf = retardation factors for the standard compounds. ETs, ellagitannins.

**Figure 3 pharmaceutics-14-02469-f003:**
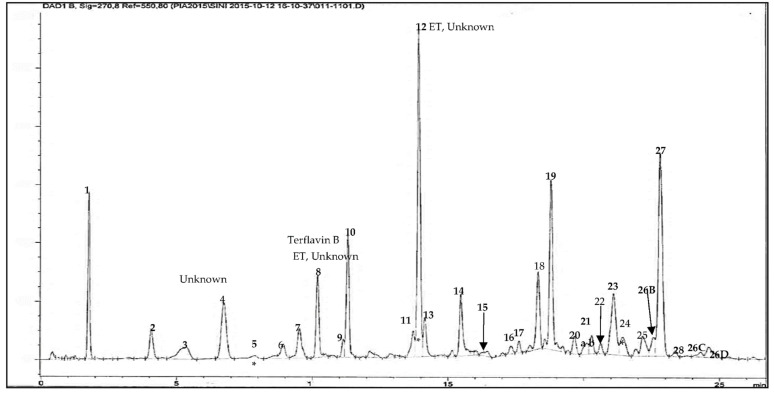
HPLC-DAD chromatogram at 270 nm of an ethyl acetate fraction of the leaves of *T. brownii*. (**1**), gallic acid; (**2**), protecatchuic acid; (**3**), unknown compound; (**4**), 1,3-di-galloyl-β-*D*-glucose; (**5**), unknown ellagitannin; (**6**), isomer of methyl-(*S*)-flavogallonate; (**7**), unknown compound; (**8**), unknown ellagitannin [M-H]^−^ 541.0291; (**9**), 1,6-di-galloyl-β-*D*-glucose; (**10**), terflavin B; (**11**), methyl gallate; (**12**), uknown ellagitannin [M-H]^−^ 617.0164; (**13**), 1,2,3-tri-*O*-galloyol-*β*-*D*-glucose; (**14**), gallotannin; (**15**), corilagin; (**16**), luteolin-7-*O*-glucoside; (**17**), gallotannin; (**18**), 3,4,6-tri-*O*-galloyol-β-*D*-glucose; (**19**), di-*O*-galloyl derivative (Maplexin D) [M-H]^−^ 467.0802; (**20**), apigenin; (**21a**), 1,2,3,6-tetra-*O*-galloyl-β-*D*-glucose; (**21b**), unknown gallotannin; (**22**), kaempferol-4’-sulfate [M-H]^−^ 365.2928; (**23**), di-methyl ellagic acid glucoside; (**24**), myricitin-3-rhmanoside; (**25**), gallotannin; (**26B**), unknown flavonoid; (**26C**), uknown flavonoid; (**26D**), unknown flavonoids; (**27**), gallotannin of chebulinic acid (Eutannin); (**28**), gallotannin; *, is the exact peak for the detected compounds.

**Figure 4 pharmaceutics-14-02469-f004:**
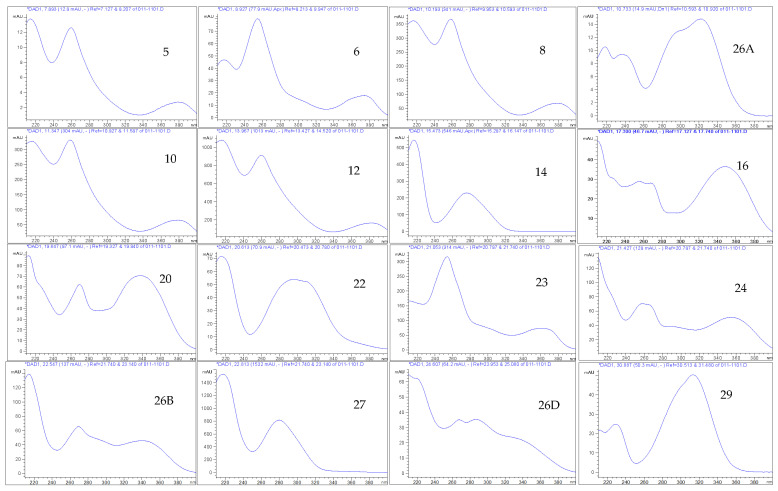
UVλ absorption maxima spectra obtained from HPLC-DAD at 270 nm for the detected compounds in *Terminalia brownii* leaf ethyl acetate extract. The used solvent for dissolving this extract is methanol to water 50%:50% (*v*/*v*). mAU (milli-Absorbance Units) in y-axis is a measure of the intensity of absorbance at various waveengths. Numbering of compound spectra according to the numbers in Figure 3.

**Figure 5 pharmaceutics-14-02469-f005:**
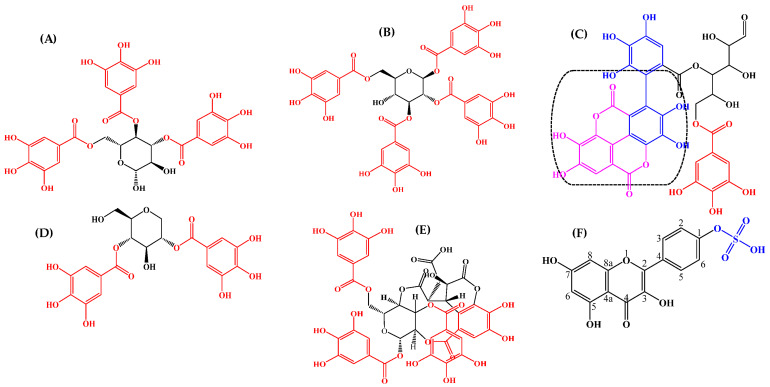
Structures of some compounds in *Terminalia brownii* leaf ethyl acetate extracts detected in this study. The compound structures were drawn according to the https://scifinder-n.cas.org/ (accessed on 10 May 2022) and https://spectrabase.com/ databases (accessed on 10 May 2022). (**A**) 3,4,6-Tri-*O*-galloyl-ß-glucopyranose; (**B**) 1,2,3,6-Tetra-*O*-galloyl-ß-*D*-glucose; (**C**) Terflavin-B; (**D**) Di-*O*-galloyl derivatives (Maplexin D); (**E**) Chebulinic acid (Eutannin); (**F**) Kaempferol-4’-sulfate. Galloyl group highlighted with the red color; hexahydroxydephinic acid (HHDP) in terflavin-B highlighted with blue color; the dashed shape is the ellagic acid part in terflavin-B.

**Table 1 pharmaceutics-14-02469-t001:** Antifungal effects of *Terminalia brownii* extracts and fractions against human pathogenic *Candida* spp. Inhibition zone diameters (IZDs) in mm.

*T. brownii* Extracts	*C. parapsilosis* ATCC 7330			*C. tropicalis* ATCC 750			*C. albicans* ATCC 10231			*C. glabrata* ATCC 2001		
	Average	SEM	AI	Average	SEM	AI	Average	SEM	AI	Average	SEM	AI
**Leaf extracts**												
L hex	15.67	0.33	0.55	** 27.00 **	0.58	1.03	NA			17.33	0.33	0.47
L Dic	16.33	0.33	0.58	16.00	0.00	0.61	16.33	0.33	0.55	** 20.33 **	0.33	0.55
L acet	** 32.93 **	0.07	1.16	** 29.83 **	0.09	1.13	** 28.00 **	0.00	0.94	** 38.60 **	0.31	1.04
L ethyl	** 31.00 **	0.00	1.09	** 24.50 **	0.29	0.93	** 28.00 **	0.00	0.94	** 22.60 **	0.21	0.61
L aqu	** 29.00 **	0.00	1.02	18.50	0.50	0.70	** 23.67 **	0.33	0.79	** 34.83 **	0.17	0.94
L HH_2_O	** 30.90 **	0.10	1.09	** 26.33 **	0.33	1.00	** 26.10 **	0.21	0.64	** 32.67 **	0.17	0.88
L H_2_O*	** 32.00 **	0.00	1.13	18.67	0.17	0.71	19.93	0.07	0.67	** 35.00 **	0.00	0.95
L Me*	** 30.23 **	0.23	1.07	** 28.80 **	0.06	1.09	** 28.27 **	0.27	0.95	** 30.67 **	0.33	0.83
L MeSox	** 31.80 **	0.06	1.12	** 30.90 **	0.06	1.17	** 29.00 **	0.00	0.97	** 32.00 **	0.00	0.86
**Stem bark extracts**												
B hex	** 20.00 **	0.00	0.71	** 23.50 **	0.29	0.89	19.33	0.33	0.65	** 25.33 **	0.33	0.68
B Dic	** 21.67 **	0.33	0.76	** 20.00 **	0.00	0.76	** 25.00 **	0.00	0.84	** 25.00 **	0.00	0.68
B ethyl	** 29.50 **	0.50	1.04	** 28.50 **	0.25	1.08	** 26.00 **	0.00	0.87	** 28.10 **	0.10	0.76
B aqu	** 26.00 **	0.00	0.92	** 20.50 **	0.29	0.78	** 20.57 **	0.30	0.69	** 21.67 **	0.33	0.59
B HH_2_O	** 28.33 **	0.33	1.00	** 26.00 **	0.00	0.99	** 23.43 **	0.30	0.79	** 28.00 **	0.00	0.76
B H_2_O*	** 27.20 **	0.20	0.96	19.50	0.29	0.74	** 23.67 **	0.33	0.79	** 22.67 **	0.33	0.61
B Me*	** 30.63 **	0.15	1.08	** 27.67 **	0.17	1.05	** 26.80 **	0.20	0.90	** 26.93 **	0.07	0.73
B MeSox	** 30.93 **	0.07	1.09	** 29.00 **	0.00	1.10	** 28.33 **	0.33	0.95	** 30.30 **	0.15	0.82
**Stem wood extracts**												
W hex	** 26.50 **	0.29	0.94	** 27.83 **	0.17	1.06	19.33	0.44	0.65	** 30.00 **	0.00	0.81
W Dic	** 25.00 **	0.00	0.88	** 24.33 **	0.33	0.92	** 26.67 **	0.33	0.89	** 25.67 **	0.33	0.69
W ethyl	** 34.67 **	0.24	1.22	15.00	0.00	0.57	** 22.50 **	0.25	0.75	** 32.60 **	0.30	0.88
W aqu	NT			19.33	0.33	0.73	NT		0.00	NT		
W HH_2_O	** 27.67 **	0.17	0.98	** 24.50 **	0.29	0.93	** 21.17 **	0.09	0.71	** 29.50 **	0.29	0.80
W H_2_O*	** 26.00 **	0.00	0.92	** 27.00 **	0.00	1.03	** 25.00 **	0.00	0.84	** 35.50 **	0.29	0.96
W Me*	18.67	0.17	0.66	18.33	0.33	0.70	** 22.33 **	0.33	0.75	15.33	0.17	0.41
W MeSox	** 25.00 **	0.00	0.88	** 20.67 **	0.67	0.78	** 24.00 **	0.00	0.80	** 34.67 **	0.33	0.94
**Root extracts**												
R hex	** 20.67 **	0.33	0.73	** 26.00 **	058	0.99	NA		NA	** 22.67 **	0.33	0.61
R Dic	** 21.00 **	0.00	0.74	** 23.00 **	0.00	0.87	** 21.67 **	0.33	0.73	** 21.00 **	0.00	0.57
R acet	** 20.50 **	0.29	0.72	** 29.67 **	0.33	1.13	** 28.00 **	0.00	0.94	** 40.67 **	0.33	1.10
R ethyl	** 21.50 **	0.29	0.76	** 30.33 **	0.33	1.15	** 26.33 **	0.33	0.88	** 29.97 **	0.03	0.81
R aqu	** 29.00 **	0.00	1.02	** 28.67 **	0.33	1.09	** 24.67 **	0.33	0.83	** 38.17 **	0.17	1.03
R HH_2_O	** 28.67 **	0.33	1.01	** 26.00 **	0.00	0.99	** 29.50 **	0.29	0.99	** 29.30 **	0.35	0.79
R H_2_O*	** 25.93 **	0.07	0.92	** 24.73 **	0.15	0.94	** 24.10 **	0.10	0.81	** 20.33 **	0.33	0.55
R Me*	** 21.50 **	0.29	0.76	** 30.00 **	0.00	1.14	** 26.33 **	0.33	0.88	** 40.00 **	0.00	1.08
R MeSox	** 31.57 **	0.13	1.11	** 31.63 **	0.19	1.20	** 29.93 **	0.03	1.00	** 32.00 **	0.00	0.86
R Ethanol wash	** 32.00 **	0.00	1.13	** 38.17 **	0.17	1.03	** 25.33 **	0.33	0.85	** 40.00 **	0.00	1.08
R Acetone wash	** 28.00 **	0.00	0.99	** 29.00 **	0.00	1.02	NT			NT		
**Root bark extracts**												
Rb hex	18.67	0.33	0.66	** 29.00 **	0	1.10	NA		NA	** 24.00 **	0.00	0.65
RbDic	19.07	0.28	0.67	19.95	0.04	0.76	19.00	0.00	0.64	** 23.00 **	0.00	0.62
Rb acet	** 31.00 **	0.00	1.09	** 22.67 **	0.33	0.86	** 27.53 **	0.29	0.92	** 33.00 **	0.00	0.89
Rb ethyl	** 31.00 **	0.00	1.09	** 31.00 **	0	1.18	** 29.90 **	0.10	1.00	** 35.50 **	0.29	0.96
Rb aqu	18.00	0.00	0.64	** 20.67 **	0.33	0.78	19.73	0.15	0.66	17.67	0.17	0.48
Rb HH_2_O	** 24.00 **	0.00	0.85	** 24.67 **	0.17	0.94	** 26.00 **	0.00	0.87	** 23.00 **	0.00	0.62
Rb H_2_O*	** 22.33 **	0.33	0.79	** 25.00 **	0.00	0.95	NT			19.67	0.17	0.53
Rb Me*	** 30.73 **	0.09	1.08	** 29.90 **	0.06	1.14	** 28.00 **	0.00	0.94	** 32.67 **	0.33	0.88
Rb MeSox	** 32.60 **	0.06	1.15	** 30.60 **	0.21	1.16	** 29.70 **	0.15	1.00	** 33.00 **	0.00	0.89
Amphotericin B	** 28.33 **	0.33	1.00	** 26.33 **	0.33	1.00	** 29.80 **	0.17	1.00	** 37.00 **	0.58	1.00

Most promising IZDs are underlined and in bold font. Acet, acetone extracts; AI, activity index; aqu, aqueous extracts; Dic, dichloromethane extracts; ethyl, ethyl acetate extracts; hex, hexane extracts; HH_2_O, hot water extracts; H_2_O*, cold water extracts; Me*, cold methanol extracts; NA, not active; NT, not tested; MeSox, methanolic Soxhlet extracts; SEM, standard error of mean of three replicates (n = 3). Ethanol and acetone washes were obtained from the root methanol Soxhlet extract using Sephadex LH-20 fractionation. Two hundred microliters of extracts at 50 mg/mL and amphotericin B at 10 mg/mL were applied to the wells.

**Table 2 pharmaceutics-14-02469-t002:** Minimum inhibitory concentration (MIC) values of *T. brownii* extracts and of some pure compounds identified in *T. brownii* extracts against *Candida* spp.

*T. brownii* extracts	*C. parapsilosis*	*C. tropicalis*	*C. albicans*	*C. glabrata*
	ATCC 7330	ATCC 750	ATCC 10231	ATCC 2001
**Leaf extracts**				
L. hex	NT	2500 *	NT	NT
L. acet	**312 ***	**625 (IC 90)**	**312 (IC 99)**	**312 (IC 99.8)**
L. ethyl	**312 (IC 93)**	1250 (IC89)	**625 (IC 94)**	2500 *
L. aqu	2500 *	NT	**312 (95.5)**	**625 (IC 97)**
L. HH_2_O	**625 ***	2500 *	**625 (IC 89.8)**	2500 (IC 99.8)
L. H_2_O*	**625 ***	NT	NT	1250 (IC 97)
L. Me*	**625 ***	1250 (IC 93.2)	1250 *	**625 (IC 95.7)**
L. MeSox	**312 (91.4)**	**625 (IC 96.2)**	1250 *	**625 (IC 98.1)**
**Stem bark extracts**				
B. hex	NT	5000 *	NT	2500 *
B. Dic	5000 *	NT	5000 *	2500 *
B. ethyl	1250 (IC 98.6)	1250 (IC 96.6)	2500 (IC 89.6)	1250 (IC 99.8)
B. aqu	2500 *	NT	NT	NT
B. HH_2_O	1250 *	2500 *	NT	1250 *
B. H_2_O*	1250 *	NT	5000 *	5000 *
B. Me*	1250 (IC 95.3)	1250 *	2500 (IC 90.6)	**312 (IC 92.4)**
B. MeSox	**625 (IC 94)**	**625 (IC 91.2)**	1250 (93.1)	**312 (IC 97.4)**
**Stem wood extracts**				
W hex	1250 *	2500 (IC 89)	NT	1250 (IC 94.1)
W Dic	2500 *	1250 *	5000 *	1250 *
W ethyl	1250 (IC 91.2)	NT	NT	**625 (IC 97.7)**
W aqu	NT	NT	NT	NT
W HH_2_O	**625 ***	5000 *	1250 (IC 99)	1250 (IC 95.4)
W H_2_O*	1250 *	2500 *	1250 *	**625 ***
W Me*	NT	NT	2500 (IC 99)	NT
W MeSox	1250 (IC 99.6)	1250 (IC 98.3)	**312 (IC 96)**	**625 (IC 98.3)**
**Root extracts**				
R hex	5000 (IC 90)	2500 *	NT	1250 (IC 98)
R Dic	5000 *	5000 *	NT	NT
R acet	5000 *	1250 (IC 93.5)	1250 *	312 (IC 96.5)
R ethyl	1250 (IC 91.3)	1250 (IC 97.5)	625 (IC 99)	625 (IC 96.5)
R aqu	1250 *	1250 *	625 (IC 97)	625 (IC 98)
R HH_2_O	1250 *	2500 *	1250 *	1250 *
R H_2_O*	2500 *	NT	NT	NT
R Me*	5000 *	2500 *	2500 *	**312 (IC 96.3)**
R MeSox	**625 (IC 92.5)**	1250 (IC 90.8)	**312 (IC 89)**	**312 (IC 95.5)**
R Ethanol wash	**93.75 (IC 97)**	**187.50 (IC 92.6)**	**187.50 (IC 975)**	NT
R Aceton wash	**375 (IC 90.5)**	NT	NT	**375 (IC 99)**
**Root bark extracts**				
Rb hex	NT	1250 *	NT	2500 (IC 93.2)
RbDic	NT	NT	NT	2500 *
Rb acet	1250 (IC 98)	5000 *	2500 *	1250 (IC 94.7)
Rb ethyl	1250 (IC 91.8)	1250 (IC 90.6)	1250 (IC 93.7)	**625 (IC 99)**
Rb HH_2_O	2500 *	2500 (IC 99)	5000 (IC 89)	5000 (IC 89)
Rb H_2_O*	5000 *	2500 (IC 95.5)	NT	NT
Rb Me*	1250 (IC 94.5)	1250 (IC 96.9)	5000 (IC 92.6)	2500 *
Rb MeSox	**625 (IC 93.3)**	1250 (IC 98.6)	2500 (IC 97.9)	1250 (IC 96.7)
**Pure compounds and amphotericin B**				
Sitosterol	**125 (IC 96.9)**	1000 (IC 99.5)	1000 (IC 93.5)	**250 (IC 96.0)**
Stigmasterol	NT	NT	1000 (IC 95.7)	500 (IC 92.4)
Gallic acid	1000 (IC 98.6)	1000 (IC 97.9)	1000 (IC 99.9)	1000 (99.8)
Ellagic acid	500 (IC 89.5)	NT	NT	500 (IC 99.5)
Quercetin	NT	NT	**250 (IC 96.6)**	**250 (95.5)**
Apigenin	125 (IC 95.5)	NT	500 (IC 97.8)	500 (IC 96.7)
Luteolin	500 (IC 99)	500 (IC 99.8)	1000 (IC 95)	NT
Corilagin	>1000	500 (IC 99.4)	>1000	NA > 1000
Friedelin	500 (IC 90)	500 (IC 91.4)	NT	NT
Amphotericin B	3.9 (IC 98.9)	7.8 (IC 93.7)	62.5 (IC 96.7)	7.8 (IC 97.5)

The smallest MIC values are indicated with bold font. Acet, acetone extracts; aqu, aqueous extracts; ethyl, ethyl acetate extracts; Dic, dichloromethane extracts; hex, hexane extracts; HH_2_O, hot water extracts; H_2_O*, cold water extracts; IC, inhibitory concentration indicating the percentage growth inhibition of the MIC concentration; Me*, cold methanol extracts; MeSox, methanolic Soxhlet extracts; NA, not active; NT, not tested. Ethanol and acetone wash obtained from a *T. brownii* Soxhlet methanol extract using Sephadex LH-20 chromatography; *, an asterisk symbol next to the numbers indicates that the results were obtained with an agar diffusion method that was used for extracts that could not be tested with the microplate method.

**Table 3 pharmaceutics-14-02469-t003:** HPLC-DAD and UHPLC/QTOF-MS data of phenols, triterpenes and polyphenolic compounds and their derivatives present in the leaf ethyl acetate extract of *Terminalia brownii*.

HPLC-DAD and UHPLC/QTOF-MS	Molecular Formula	tR HPLC-DAD (min)	tR UHPLC/QTOF-MS (min)	[M-H]^−^ (*m*/*z*)	Exact Mass (calc.)	UV_λ_ Absorption Max. from HPLC-DAD	Peak Area (%) at 270 nm	PPM Value
Sample and Compounds								
**Gallic acid (1)**	C_7_H_6_O_5_	1.760	1.165	169.0146	170.0213	216, 272	5.55	6.47
gallotanin		2.074				218, 266	0.06	
ellagitannin		3.251				216, 260, 380	0.02	
**Protocatechuic acid (PCA) (2)**	C_7_H_6_O_4_	4.076	2.347	153.0198	154.0264	216, 220, 260, 294	1.79	7.79
gallotannin		4.488				222, 288	0.10	
Unknown compound **(3)**		5.351				216	1.60	
**1,3-di-galloyl-β-*D*-glucose (4)**	C_20_H_20_O_14_	6.734	4.248	483.0796	484.0846	216, 272	4.65	5.78
ellagitannin **(5)**		7.896				214, 260, 378	0.41	
gallo-ellagitannin		8.633				222, 258, 376	0.29	
**Isomer of methyl-(*S*)-flavogallonate (6)**		8.925	8.52	483.0811		218, 254, 374	1.06	
Unknown compound **(7)**		9.540				220	2.99	
ellagitannin **(8)**		10.199	3.363	541.0291		216, 258, 380	3.91	
ellagitannin		10.490				218, 258, 364	0.13	
**Unknown flavonoid (26A)**		10.708				218, 234, 298, 322	0.12	
**1,6-di-galloyl-β-*D*-glucose (9)**	C_20_H_20_O_14_	11.142	4.714	483.0808	484.0846	216, 272	0.68	8.26
**Terflavin B (10)**	C_34_H_24_O_22_	11.303	3.546	783.0701	784.0750	216, 258, 378	5.37	3.70
gallotannin		11.481				216, 256	0.09	
ellagitannin		12.114				220, 256, 378	0.43	
ellagitannin		12.867				214, 260, 380	0.28	
ellagtannin		13.114				214, 260, 378	0.12	
gallotannin		13.300				218, 278	0.06	
**Methyl galate (11)** (gallic acid based) (Methyl 3,4,5-trihydroxybenzoate)	C_8_H_8_O_5_	13.722	4.013	183.0304	184.0369	216, 280	1.52	7.06
ellagitannin (unknown) **(12)** {main compound}		13.922	4.529	617.0164		216, 260, 382	15.46	
**1,2,3-tri-*O*-galloyol-β-*D*-glucose (13)**	C_27_H_24_O_18_	14.146	5.680	635.0889	636.0954	218, 272	1.98	2.04
gallotannin		15.154				216, 272	0.29	
gallotannin **(14)**		15.473				216, 276	3.32	
gallotannin		15.722				216, 258, 380	0.30	
gallotannin		15.891				218, 274	0.19	
**Corilagin (15)**	C_27_H_22_O_18_	16.031	5.545	633.0756	634.0798	216, 266, 354	0.20	5.68
gallotannin		16.450				216, 292	0.28	
gallotannin		17.011				218, 278	0.13	
**Luteolin-7-*O*-glucoside (16)**	C_21_H_20_O_11_	17.317		447.0945	448.0999	228, 256, 268, 350	0.46	5.36
gallotannin **(17)**		17.611				216, 276	0.53	
ellagitannin		17.812				218, 256, 368	0.01	
gallotannin		18.013				220, 276	0.28	
**3,4,6-tri-*O*-galloyol-β-*D*-glucose (18)**	C_27_H_24_O_18_	18.312	5.661	635.0904	636.0954	224, 276	4.07	4.40
galloyl ellagitannin		18.563				218, 274, 382	0.38	
**gallotannin (19)** (Di-*O*-galloyl derivative) (Maplexin D)	C_20_H_20_O_13_	18.785	6.094	467.0802	468.0897	218, 256	8.31	−3.63
gallotannin		18.967				216, 278	0.41	
galloyl ellagitannin		19.225				216, 276, 342	0.26	
**Apigenin (20)**	C_15_H_10_O_5_	19.651		269.0455	270.0525	212, 270, 338	0.98	2.96
**1,2,3,6-tetra-*O*-galloyl-β-*D*-glucose (21a)**	C_34_H_28_O_22_	20.113	6.572	787.0872	788.1062	218, 279	0.88	−14.21
unknown gallotannin **(21b)**		20.288				218, 274	1.04	
**Kaempferol-4′-sulfate (22)**	C_15_H_10_O_9_S	20.617	7.010	365.2928	3663	216, 298, 308	0.50	1.64
**Di-methyl ellagic acid glucoside (23)**	C_22_H_20_O_13_	21.089	7.750	491.0843	492.0897	210, 254, 366	5.08	4.88
**Myricetin-3-rhamnoside (24)**	C_21_H_20_O_12_	21.422	7.800	463.3798	464.3800	210, 258, 354	1.70	16.37
gallotannin		21.909				218, 282	0.41	
gallotannin **(25)**		22.158				216, 280	1.71	
Unknown flavonoids **(26B)**		22.557				214, 270, 340	1.19	
**Chebulinic acid (Eutannin) (27)** or (1,3,6-tri-*O*-galloyl-2,4-*O*-chebuloyl-*β*-*D*-Glc)	C_41_H_32_O_27_	22.800	8.159	955.0790	95.608	218, 280, 381	13.33	7.11
gallotannin **(28)**		23.362				218, 282	0.33	
Unknown flavonoids **(26C)**		24.290				214, 264, 286, 350	0.31	
Unknown flavonoids **(26D)**		24.602				210, 222, 268, 286	1.50	
gallotannin		26.253				216, 282	0.14	
gallotannin		29.000				216, 280	0.17	
**Genipin (29)**	C_11_H_14_O_5_	30.878	12.222	225.1504	225.9037	210, 233, 288	0.33	
**Sericic acid (30)**	C_30_H_48_O_6_	35.440	13.937	503.3379	504.3438	210, 290, 328	0.03	3.77
**ellagic acid derivative (31)**		36.426				212, 250, 306, 370	0.04	
Unknown compound **(32)**		38.726	17.834	293.1759			0.05	
**Genkwanin (Apigenin 7-*O*-methyl ether) (33)**	C_16_H_12_O_5_	39.012	18.417	283.192	284.1981	210, 218, 288	0.05	5.98

The exact calculated mass (M+) was obtained from the molecular formula and is the sum of the exact masses and numbers of the atoms in the molecule; UV_λ_ absorption maxima for identified compounds were obtained from HPLC-DAD at 270 nm. Peak area % was obtained from an HPLC-DAD chromatogram at 270 nm, and PPM is mass errors calculated as parts per million for measuring the accuracy of 5 ppm (±2 ppm precision). The bold text indicates the most important compounds that were identified and discussed in this article.

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
