# Peer review of "Anti-Candida Activity of Extracts Containing Ellagitannins, Triterpenes and Flavonoids of Terminalia brownii, a Medicinal Plant Growing in Semi-Arid and Savannah Woodland in Sudan"

_pharmaceutics, 2022, doi:10.3390/pharmaceutics14112469_

Round 1
Reviewer 1 Report
The work is original, professional, with detailed explanations of both the methods that were applied and the results. Experimental research is abundantly and thoroughly explained through various forms, tabelar and histogram. I accepted the paper in the form you sent me.
Author Response
Author’s Reply to the Review Report (Reviewer 1):
Thank you for your opinion to accept our manuscript as it is, although we agree with the reviewer No. 3 to correct many places in the manuscript before acceptance. We have made English language spell checking according to your wishes and these changes are highlighted with the red color in the manuscript. Thank you for accepting the paper for publication in Pharmaceutics.

Reviewer 2 Report
Candida spp. are the most common cause of fungal infection in critically ill patients, and the emergence of resistant Candida spp. against the currently available antifungal drugs prompt researchers to develop new antifungal agents, from natural sources. The work of Saleh et al. explores the potential of brownii extracts as antifungal by investigating its anti-Candida activity and phytochemistry. Overall the work is well organized and presented. It contains a huge quantity of results and experimental work. Analytical methods and techniques used to characterize the pant-extracts are consistent with the literature and corroborate the objectives. The results presented are coherent and their presentation, interpretation, and conclusions are supported by literature data. The experiment section: (2. Materials and Methods) is exhaustive. It is well written, clear, and all the experiments are described in a lot of details which is good and supports reproducibility. However some details are too much, for instance, page 5 line 183, sterile filter paper disks 12.7 mm, Schleicher and Schuell 2668), too much detail. The same remark for page 6 line 227, "Using a UV-Visible 227 Spectrophotometer (Pharmacia LKB-Biochrom 4060)". The spectrophotometer used has been described in page 5 line 183.No need to re-describe it. UVλ-visible spectrophotometer ? please delete lambda .
This section or at least a part of this section can be moved to supplement information.
The legend of Figure 5. is not clear and is confusing, please make it clear
"Uv-visible absorption spectra recorded from...." in Y-axis what does mean mAU? mili arbitrary unit?
Regarding the polarity of the solvents and its effect on extraction yield, antifungal effects and fractions against human pathogenic, etc. (table 1 & 2). Ethyl acetate is less polar than dichlormethane, however the screening of phytochemical activities do not follow a clean polarity solvent trend. Please make a comment or make it clear.
Author Response
Authors Reply to the Review Report (Reviewer 2):
Point 1, We thank reviewer No 2, for praising our work to be well organized, and presented and corroborated to the objectives of the manuscript. We disagree with the reviewer that the material and method part has too many details. We would like to keep the details about the laboratory material that was used for the experiments, such as the sterile filter paper disks (Æ 12.7 mm, Schleicher and Schuell 2668) and UV-Visible 227 Spectrophotometer (Pharmacia LKB-Biochrom 4060)" in page 5 line 183 (now moved to line 185 and 186). We think that the data on the material that was used is important scientific information for this kind of experiments to be repeated. Besides, we have no supplement part with the laboratory material described in detail in this manuscript. Also, we would like to keep the details about Spectrophotometer in page 6 line 227 (now page 6 Line 229) since it is described here to be used for the preparation of the inoculum for a turbidimetric microplate method. We agree to delete the lambda in UVl-visible spectrophotometer in Line 178.
Point 2. We have tried to improve the resolution in Figure 4 and the details about the mAU (milli-Absorbance Units) in y-axis was added to Figure 4 caption and highlighted with the red color.
Point 3: We disagree with the statement that the ethyl acetate is less polar than dichloromethane. According to the polarity index which is defined as a measure of the ability of the solvent to interact with various test solutes, the polarity index for dicholoromethane is 3,1 and ethyl acetate have polarity index of 4.4, thus ethyl acetate is more polar than dicholoromethane. Besides, screenings using solvents of various polarities gives a better chance to find the part of the plant material (polar, intermediate or non-polar compounds) that is responsible for the good antifungal activity.

Reviewer 3 Report
Please see the attached file for comments.

Author Response
Author’s Reply to the Review Report (Reviewer 3):
Point 1, We thank reviewer No 3 for the valuable comments. An English language spell check was done according to the reviewers wish. Moreover, the results are more clearly presented and the text that has been changed is highlighted with the red color in this manuscript.
We have corrected the Figure numbering and citation through the whole manuscript, and all changes were marked and highlighted with the red color. In addition, Figure 5, that is a new figure added to this work, is cited in the manuscript.
Specific comments:
- Line 37: It should be “diseases; HPLC-DAD” instead of “diseases HPLC-DAD”. Our response: We have placed the semi-colon; between diseases and HPLC-DAD.
- Lines 23, 38, 86, 295, 301, 315, 508, and Table 3 (title and column headings): The acronym for ultra- high performance liquid chromatography-quadrupole time-of-flight mass spectrometry should be unified. Our response: We have now unified the HPLC-DAD and UHPLC/QTOF-MS in lines 23, 38, 86 as well as in lines 295, 301,315, 508 that have become Lines 300, 306, 320 and 513 respectively. The acronymns, HPLC-DAD and UHPLC/QTOF-MS have been unified also in the Table 3 title and in the column headings of that table Lines 544 and 546.
- Line 45: Italic font should not be used for “spp.”. Our response: We agree with the reviewer comments. We have corrected the abbreviation spp. throughout the manuscript from italic to normal font.
- Line 55: It should be " [7,9–11].” Instead of “[7], [9–11].” Our response: In line 55 we have changed the citation from “[7], [9–11]” to " [7,9–11].”
- Line 79: It should be” [17,20,24,26–29]” instead of “[17,20,24], [26–29]”. Our response: The references are now cited as “[17,20,24,26–29]” and highlighted with red color in Line 79.
- Line 86: It should be “UHPLC“ Instead of “UPLC“. Our response: we agree with the reviewer, we have changed the “UPLC“ to “UHPLC“ throughout the whole manuscript.
- Lines 109, 139, 142, 227, 239: An appropriate value of relative centrifugal force (RCF) should be given instead of Our response: we disagree with this comment, we think that both rpm (rotation per minute) and RCF (Relative centrifugal force) values could be used to describe the used centrifugation spinning value. However, both value (rpm and RCF) could be converted to each other and vice versa through the below formula:
RCF = 11.2 × r (RPM/1000)2 or RCF = 1.12 × 10-5 (RPM)2.
- Line 167: Italic font should not be used for “ATCC 10231”. Our response: Italic font for “ATCC 10231” is not in Italic font now in Line 167.
- Line 168, Table 1 and 2: Strain number of parapsilosis ATCC 7330 (line 168) is not consistent with C. parapsilosis ATCC 22019 given in Table 1 and Table 2. This issue should be addressed. Our response: we have corrected the ATCC number for C. parapsilosis in Table 2 and it has been changed from C. parapsilosis ATCC 22019 to the C. parapsilosis strain of ATCC 7330 that has been used in all our experiments for this paper.
- Line 179: It should be “Petri dishes“ instead of “petri dishes”. Our response: Petri dishes now has been correct from small starting letter of “petri dishes“ to capital letter in “Petri dishes“ and now in Line 181.
- Line 181: Microliters should be uniformly Our response: “µL“ has been changed to “µl“ and moved now to the Line 183.
- Lines 218-219 and 265-266: Ellagic acid source should be unified. Our response: The supplier of ellagic acid was corrected to be (E-2250, Sigma-Aldrich, England) both in line 220 and in Line 270 respectively.
- Line 231: The acronym “GT” should be defined more Our response: We have explained the GT in detail in Lines 232-235: as ``Moreover, the test wells (GT) contained 100 μl of the two-fold diluted extracts, fractions, compounds or antibiotics and 100 μl of the diluted fungal suspension, thus also containing 5 ´ 105 CFU/ml in the microplate wells
- Line 260: It should be “Figure 1” instead of “Figure 2” (see also comment no. 19). The font color should be changed from green to Our response: Thank you for noticing that the text in line 260 refers to Figure 1. We have changed it now. The green font in Figure 1 was changed to red in Line 265.
- Line 273: “Camaq”? or “Camaq”? Our response: “Camaq” in line 273 corrected to “Camaq” now in Line 278.
- Line 274: It should be “366 and 254 nm” instead of “336 and 254 nm”. Our response: the error of “366 ” nm in line 274 have been corrected to “366 ” and now in Line 279.
- Line 295: An extra space should be Our response: The extra spaces have been deleted
- Line 327: It should be “Figure 1“instead of “Figure 2”. This is the first figure in the Our response: “Figure 2“in Line 327 corrected to “Figure 1“and now in Line 332. Also the figures number corrected in the whole manus accordingly.
- Line 333: The values “28‒36.5 mm” are not consistent with those given in Table 1 for the acetone extract of the It should be “28-38.6 mm”. Our response: we agree with the reviewer “28‒36.5” in line 333 corrected to “28-38.6 mm” and now in Line 338.
- Line 336: The value “27.00” is not consistent with that given in Table 1 for the acetone extract. It should be “29.83”. Our response: we have corrected the IZD in Line 336 from “27.00” to “29.83” and now in Line 341.
- Line 372: It should be “growth” instead of “Growth “. Our response: the word of “Growth “in Line 372 have been changed to “growth “and now in Line 373.
- Line 383: It should be “Figure 1“instead of “Figure 2“(see also comment no. 19). Besides, a redundant punctuation mark (period) should be deleted. Our response: The redundant punctuation has been deleted and we have corrected Figure 2 to Figure 1 in Line 383 (now in Line 388 in the manuscript).
- Line 386: It should be “Figure 1“instead of “Figure 2“(see also comment 19). Our response: We have corrected now “Figure 2“in Line 386 and become “Figure 1“in Line 391.
- Table 1: See comment 10 for C. parapsilosis strain. Our response: the ATCC number of Candida parapsilosis has now been corrected in Table 1.
- Table 1: I suggest a separate row for “Amphotericin B“. Besides, the spelling of amphotericin should be standardized throughout the I suggest “amphotericin B” instead of “amphotericin-B”. Our response: We have now marked amphotericin B on table 1 so that it is on a separate row. We also agree to changing the way of writing amphotericin-B to amphotericin B. Now we have changed to amphotericin B in whole manuscript.
- Lines 403-405: I suggest an alphabetic order for abbreviations. The abbreviations “NA” and “NT” used in Table 1 should be defined. Our response: We agreed with reviewer comments in Line 403-405, we have alphabetically ordered the abbreviations now in Table 1 captions, now in Lines 408-409. Also, we have defined the abbreviations of NA and NT in the caption of Table 1, so that NT means not tested and NA stands for not active.
- Lines 404 and 409: It should be “Soxhlet“instead of “soxhlet“. Our response: In Line 404 we have corrected “soxhlet“to “Soxhlet“ and now in Line 409. Also, In Line 409 we have corrected “soxhlet“to “Soxhlet“ and now in Line 415
- Lines 408-410: I suggest an alphabetic order for abbreviations. The abbreviations “NA” and “NT” used in Table 2 as well as asterisk symbol next to numbers should be Our response: We agree with this reviewer comments, we have alphabetically ordered the abbreviation in Table 2 captions as “acet, acetone extracts; aqu, aqueous extracts; ethyl, ethyl acetate extracts; Dic, dichloromethane extracts; hex, hexane extracts; HH2O, hot water extracts; H2O*, cold water extracts; IC, inhibitory concentration indicating the percentage growth inhibition of the MIC concentration; Me*, cold methanol extracts; MeSox, methanolic Soxhlet extracts“ and now in Line 413-416, also we have defined the asterisk symbol as “ *, An asterisk symbol next to the numbers indicates that the results were obtained with an agar diffusion method. We have also defined the abbreviations of NA and NT as in Table 1; as NT for not tested and NA for not active “The caption text is now in Lines 413-416.
- Lines 425-432: “Figure 2” should be renamed to “Figure 1“(see also comment 19). Our response: We have corrected now “Figure 2“in Line 425-432 and now it is “Figure 1“in Line 437 “The caption Figure 1”. Also, we have inserted “Figure 2“in Line 446 and 447.
- Line 437: It should be “Figure 2“instead of “Figure 3“(See comment no.30. Besides, Figure 3 was not included in the manuscript). Our response: we agree with the reviewer comments and now the word “Figure 3“has been deleted and instead “Figure 2“was placed in Line 442. Moreover, the other Figure numbers in the manuscript have been changed so that Figure 4 changed to Figure 3 and so on. The corrected Figure numbers have also been changed in the text.
- Line 438: It should be “254 and 366 nm“instead of “256 and 366 nm“. Our response: we have corrected the wavelength number form “256 and 366 nm“to “254 and 366 nm“ now in Line 443.
- Lines 443-447: “Figure 1” should be renamed to “Figure 2” (see comments 30 and 31). Besides, there is no “E” panel in the figure. Our response: We have corrected “Figure 1” Lines 443-447 and renamed to “Figure 2 as well as the “E” panel in the “Figure 2” have been placed and now in Lines 452-453.
- Line 453: It should be “Figure 3“instead of “Figure 4“(see also comments 19, 30, 31.). Our response: we changed “Figure 4“ in line 453 to “Figure 3“and now in Line 458.
- Line 455: It should be “[39]). “instead of “[39].“. Our response: We agree with the reviewer we have placed bracket after the reference number of “[39]) and now in Line 460.
- Line 463: It should be “Figure 3 and 4“instead of “Figure 4 and 5“ (see comment 34). Our response: we agree with reviewer we have changed “Figure 4 and 5“in Line 463 to “Figure 3, 4 and 5 “ and now line 468.
- Lines 464, 467, 470: The abbreviation “UPLC-DAD” is not Our response: we have corrected the “UPLC-DAD” in line 464, 467,470 to “UHPLC-DAD”, now line 469-476, we did not think that the abbreviation of “UPLC-DAD” needs to be defined here again since according to the journal guideline it have to be define once, in the first time where it have been appeared.
- Lines 464-651 and Table 3: Two different abbreviations are used for “retention times”: “Rt” in Table 3 and “tR” in the This issue should be addressed. Our response: we have unified the “retention times” abbreviation as“ tR” in Table 3 and the abbreviation of “Rt” have been deleted from this manuscript.
- Line 476: It should be “Figure 3“instead of “Figure 4 “. Our response: We agree with reviewer and now “Figure 4“in Line 476 corrected to “Figure 3“ and become in Line 493.
- Line 482: “(m/z 483.0665)” seems to be incorrect. For 1,6-di-galloyl-β-D-glucose (9) m/z 483.0808 is given in Table Our response: the m/z for the 1,6-di-galloyl-β-D-glucose in Line 482 have been corrected to “m/z 483.0808”, now in line 486.
- Lines 484-485: Values (m/z 0723) for 1,2,3-tri-О-galloyol-β-D-glucose (13) and (m/z 635.07399) for 3,4,6-tri-О-galloyl-β-D-glucose (18) should be verified (see Table 3). Our response: The m/z for 1,2,3-tri-О-galloyol-β-D-glucose (13) and 3,4,6-tri-О-galloyl-β-D-glucose (18) were corrected in the manuscript, now in Line 487-490.
- Line 488: It should be “Figure 3“ instead of “Figure 4“. Our response: we have corrected “Figure 4“ to “Figure 3“ and now in line 493.
- Lines 513-538. “Figure 4” should be renamed to “Figure 3“and “Figure 5” to “Figure 4” (see also comments 36, 39 and 42}. Our response: we have corrected “Figure 4“to “Figure 3“ and “Figure 5” is renamed to “Figure 4” now in Line 542-543.
- Line 538: It should be “numbers in Figure 3.“ instead of “numbers in Figure “. Our response: We have renamed “numbers in Figure 4.“ to “numbers in Figure 3.“.
- Table 3 (title and column headings): see comment 3. Our response: We agree with the reviewer, and now we have unified the HPLC-DAD and UHPLC/QTOF-MS in Table 3 (title and column headings), see our comments in point number 3.
- Table 3: I suggest to add “(m/z)” to the column [M-H]-. Our response: Thank you for this suggestion, we have accepted your suggestion and inserted “(m/z)” in Table 3 with column heading of [M-H]-.
- Lines 571-572 and 650: Citations should be Our response: The citation of reference numbers “[17,21,22,29,38], [46‒50]” in line 571-572, have been merged, now in Line 577. also reference “[39], [60‒62]” in Line 650 have been merged with the other reference as “[39,60‒62]”, now in the line 656-657.
- Line 587: It should be “Figure 3“ instead of “Figure 4”. Our response: we agree with the reviewer. In Line 587 we have corrected “Figure 4“to “Figure 3” now in Line 593.
- Line 615: It should be “Figure 1“instead of “Figure 2”. Our response: we agree with the reviewer, in Line 615 we have corrected “Figure 1“to “Figure 2“, now in Line 621.
- Line 622: It should be “Figure 2“instead of “Figure 3”. Our response: we agree with the reviewer, in Line 622 we have changed “Figure 3” to “Figure 2“, now in Line 628.
- Line 634: It should be “[58,59].“ instead of “[58.59].”. Our response: we agree with the reviewer, in Line 634 the reference numbers “[58.59].” have been corrected to “[58,59].“ and now in Line 640.
- Line 644: It should be “Figure 3“instead of “Figure 4”. Our response: In line 644 we have corrected “Figure 4” to “Figure 3”, now in in Line 650.
- Line 645: Spelling of terflavin B should be unified throughout the Our response: We have unified “terflavin B“and “terflavin-B“ throughout the manuscript to appear as “terflavin B“.
- Lines 697 and 758: The font color should be changed from green to Our response: In line 697 we have changed the color of “Table 3“ from green to red and now in line 705, as well as in line 758 we have changed the color of “Table 2“ from green to red and now in line 767.
- Line 711: Italic font should not be Our response: we agree with reviewer in line 711 “protocatechuic acid“ have been changed to “protocatechuic“´in the subtitle in line 719.
- Lines 716, 751, 761, 791: It should be “et ” instead of “et al”. Our response: we agree with reviewer, in Lines 716, 751, 761, and 791 “et al” have been corrected with the dot to “et al.” and the dot were thoroughly highlighted with the red color, the changes now in Lines 724, 749, 760 and 770 respectively, also other places were corrected in Lines 672,690, 692, 802 and 811.
- Line 732: Italic font should be used for species Our response: We agree with the reviewer, in line 732, the plant species names of in “Frankenia laevis and Argyreia speciose” have been written in Italic font as “Frankenia laevis and Argyreia speciosa” and now in line number 741.
- Line 739: It should be “ albicans” instead of “C.albicans”. Our response: we agree with these comments, in Line 739 dot have been added to the Candida abbreviation “C. albicans”, now in Line 748.
- Line 781: It should be “[96,98,99]” instead of “[[96,98,99]”. Our response: We agree with the reviewer, in line 781 the extra bracket of “[[96,98,99]”have been deleted and now the corrected ones of “[96,98,99]” located in Line 791.
- Line 830: It should be “O.L. contributed” instead of “O.L contributed”. Our response: we agree with the reviewer, in line 830 dot was added to “O.L contributed” and now have been changed to “O.L. contributed” in line 842.
- References should be verified (See comment no. 1). Besides, italic font should be used for species Our response: We agree with the reviewer, the references have been verified through the whole manuscript and the species name and scientific names we placed in italic font in line 891, 937, 941-942 and 954-955.

Round 2
Reviewer 3 Report
The revised version of the manuscript ““Anti-Candida activity of extracts, containing ellagitannins, triterpenes and flavonoids of Terminalia brownii, a medicinal plant growing in Semi-Arid and Savannah Woodland in Sudan” has been sufficiently improved to warrant publication in Pharmaceutics.
However, I suggest to verify Table 2 caption of the revised manuscript (see line 415 and compare it with the answer to comment no. 29 regarding an asterisk symbol).